# Increased future ocean heat uptake constrained by Antarctic sea ice extent

Linus Vogt[1,2,5], Casimir de Lavergne[1], Jean-Baptiste Sallée[1], Lester Kwiatkowski[1], Thomas L. Frölicher[3,4], and Jens Terhaar[2,3,4]

[1]Sorbonne Université, CNRS/IRD/MNHN, Laboratoire d'Océanographie et du Climat Expérimentations et Approches Numériques (LOCEAN), Paris, France
[2]Woods Hole Oceanographic Institution, Woods Hole, MA, USA
[3]Climate and Environmental Physics, Physics Institute, University of Bern, Bern, Switzerland
[4]Oeschger Centre for Climate Change Research, University of Bern, Bern, Switzerland
[5]Courant Institute of Mathematical Sciences, New York University, New York, New York, NY, USA

**Correspondence:** Linus Vogt (linus.vogt@nyu.edu)

**Abstract.** The ocean takes up over 90% of the excess heat stored in the Earth system as a result of anthropogenic climate change, which has led to sea level rise and an intensification of marine extreme events. However, despite their importance for informing climate policy, future ocean heat uptake (OHU) projections still strongly differ between climate models. Here, we provide improved global OHU projections by identifying a relationship between present-day Antarctic sea ice extent and future OHU across an ensemble of 28 state-of-the-art climate models. Models with more sea ice at present also simulate a colder Southern Hemisphere climate state in general, allowing for a larger shift in atmospheric and ocean warming. This regional change affects global warming and heat uptake via a northward propagating cloud feedback. Combining this relationship between historical Antarctic sea ice extent and future global OHU with satellite observations of Antarctic sea ice reduces the uncertainty of OHU projections under future emissions scenarios by 12–33%. Moreover, we show that an underestimation of present-day Antarctic sea ice in the latest generation of climate models results in an underestimation of future OHU by 3–14%, of global cloud feedback by 19–32%, and of global atmospheric warming by 6–7%. This emergent constraint is based on a strong coupling between Antarctic sea ice, deep ocean temperatures, and Southern Hemisphere sea surface temperatures and cloud cover in climate models. Our study reveals how the present-day Southern Ocean state impacts future climate change, and suggests that previous constraints based on warming trends over recent decades have underestimated future warming and ocean heat uptake.

## 1  Introduction

Since the beginning of the industrial period, the ocean has taken up over 90% of the excess heat generated by human-caused climate change (Forster et al., 2021). This ocean heat uptake (OHU) has limited the rate of atmospheric temperature increase (Liu et al., 2016), but the widespread warming of the ocean (Johnson and Lyman, 2020) has had cascading negative consequences for humans and marine ecosystems. Ocean warming contributes to sea level rise through thermal expansion and the melting of marine-terminating glaciers (Cazenave and Llovel, 2010). Sea level rise and ocean warming create risks for coastal

communities due to increased flooding and more destructive tropical cyclones (Sun et al., 2017; Pörtner et al., 2022). Higher upper ocean temperatures also lead to changes in stratification and the supply of nutrients and oxygen to marine ecosystems (Sallée et al., 2021; Bopp et al., 2013; Morée et al., 2023), impacting fish stocks (Cheung et al., 2016) and perturbing the global carbon cycle (Joos et al., 1999; McNeil and Matear, 2013). Furthermore, ocean warming drives more frequent and intense marine heatwaves, potentially causing widespread collapses of foundation species including corals, kelps, and seagrasses (Frölicher et al., 2018; Smith et al., 2023).

These direct negative impacts of ocean warming imply a need for accurate projections of OHU under future climate change. The magnitude of future OHU primarily depends on cumulative greenhouse gas emissions, and thus on the effectiveness of mitigation policies (Fox-Kemper et al., 2021). However, for any given level of greenhouse gas emissions, OHU is also influenced by the strength of climate feedbacks as well as oceanic ventilation and overturning (Zelinka et al., 2020; Marshall et al., 2015). Climate feedbacks such as cloud and albedo feedbacks alter the radiative balance of the Earth and thus affect the transient climate response, climate sensitivity, and future ocean heat storage (Zelinka et al., 2020; Williams et al., 2020). In turn, the efficiency at which the ocean transports heat from the surface layer to the deep ocean influences its capacity for heat storage and can modulate climate feedbacks by affecting surface warming patterns (Winton et al., 2010; Armour et al., 2013; Andrews et al., 2022).

Regionally, the majority of OHU occurs in the Southern Ocean (Frölicher et al., 2015). In an observation-based reconstruction, the Southern Ocean south of roughly 40°S accounts for around 67% of global OHU between 1871 and 2017 (Zanna et al., 2019). In climate model simulations from phase 6 of the Coupled Model Intercomparison Project (CMIP6; Methods), the Southern Ocean south of 30°S is responsible for 84% (68–99%) of the global historical OHU from 1850 to 2024, 53% (38–62%) of future OHU from 2024 to 2100 under the low-emissions SSP1-2.6 scenario, and 48% (42–52%) under the high-emissions SSP5-8.5 scenario (inter-model uncertainty is expressed as 66% *likely* ranges) (Frölicher et al., 2015; Shi et al., 2018). The disproportionately large heat uptake in the Southern Ocean is a direct consequence of the vigorous deep-reaching overturning in this region (Armour et al., 2016). The overturning in the high-latitude Southern Ocean is driven by strong westerly winds which provoke upwelling of large volumes of cold water from the deep ocean (Marshall and Speer, 2012). Much of this upwelled water is warmed by the atmosphere before being subducted back into the ocean interior further northward as mode and intermediate waters, following the upper cell of the Southern Ocean meridional overturning circulation (Armour et al., 2016; Sallée, 2018; Talley, 2013).

Although robust and precise projections of OHU are paramount for informing climate mitigation and adaptation measures, accurately projecting OHU remains challenging (Cheng et al., 2022) (Fig. 1). The uncertainty of the future cumulative global OHU from 2024 to 2100 is 23–28% of the multi-model mean (depending on emissions scenario), and the ranges of cumulative OHU projections for 2100 overlap across scenarios (Fig. 1). Uncertainties in future OHU are large because cloud feedbacks and oceanic heat sequestration by ocean ventilation and mixing remain notoriously challenging to correctly simulate (Frölicher et al., 2015; Ceppi et al., 2017; Zelinka et al., 2020; Terhaar et al., 2021). The Southern Ocean overturning is particularly difficult to faithfully simulate in Earth system models (ESMs) such as those participating in CMIP6 (Beadling et al., 2020). Biases in the baseline state of ESMs are known to have global repercussions on projected climate change, notably by preconditioning

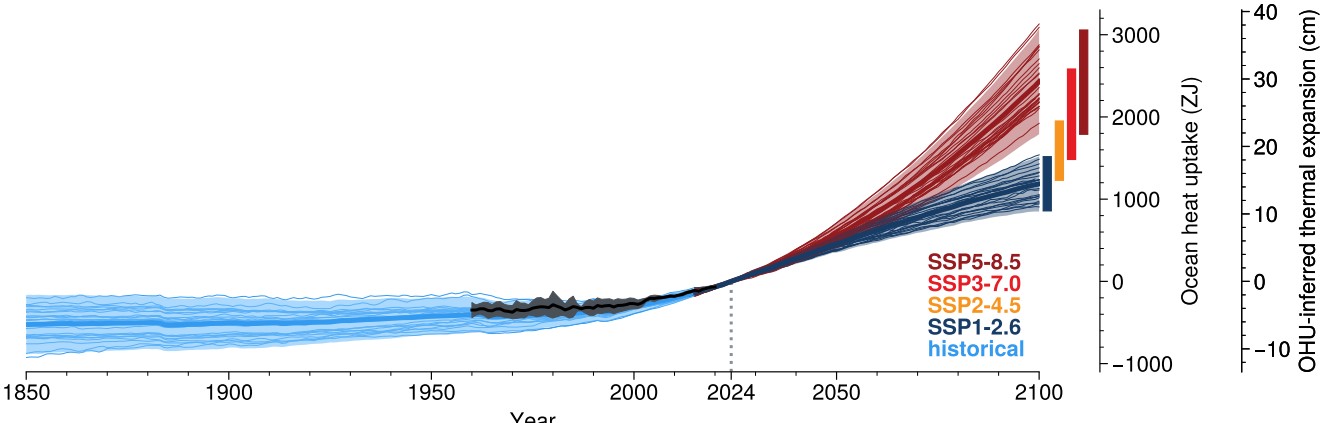

**Figure 1. Ocean heat uptake in CMIP6 models.** Globally integrated cumulative historical and future ocean heat uptake relative to the year 2024 under different scenarios and the associated global mean sea level rise through thermal expansion (see Methods). Thin lines are individual models, while the thick lines and shading depict respectively the ensemble mean and standard deviation for each scenario. The coloured bars on the right indicate the 95% confidence interval around the mean OHU in 2100 for each scenario. Coloured bars are shown for four scenarios (SSP5-8.5 in dark red, SSP3-7.0 in bright red, SSP2-4.5 in orange, SSP1-2.6 in dark blue), whereas the time series are shown only for the historical period and the SSP1-2.6 and SSP5-8.5 scenarios. The black curve and grey shading show the observed changes over 1960–2020 (Minière et al., 2023).

future cloud feedback (Kajtar et al., 2021; Shin et al., 2023; Siler et al., 2018) and Southern Ocean ventilation (Terhaar et al., 2021; Bourgeois et al., 2022).

One approach to reducing inter-model uncertainties is the method of emergent constraints (Hall and Qu, 2006; Hall et al., 2019; Eyring et al., 2019). An emergent constraint identifies a physically grounded relationship between an observable historical climate variable and an uncertain future climate variable, across an ensemble of models. Combining this quantitative relationship with observations of the historical variable yields a constrained estimate of the uncertain future variable. Emergent constraints can be broadly divided into three categories (Sanderson et al., 2021): (i) trend-on-trend constraints that assume a time invariant model bias that has existed over the historical period and will continue in the future (e.g., Tokarska et al., 2020; Lyu et al., 2021; Nijsse et al., 2020; Jiménez-de-la Cuesta and Mauritsen, 2019), (ii) process-based constraints that identify a physical or biochemical bias that causes a mechanistically linked bias in projections of the considered variable (e.g., Terhaar et al., 2021, 2020; Bourgeois et al., 2022), and (iii) sensitivity-based constraints where the sensitivity of a system to changes on short timescales, such as seasonal changes, is related to the response of a system to climate change (e.g., Hall and Qu, 2006; Kwiatkowski et al., 2017). Trend-on-trend constraints have previously indicated smaller future OHU and atmospheric warming compared to the unconstrained mean of CMIP6 projections (Lyu et al., 2021; Tokarska et al., 2020). However, trend-on-trend constraints can fail if the historically-observed trend is not representative of a time invariant bias. This can occur either because the past trend has been strongly affected by a particular phase of natural variability (England et al., 2014; Marvel et al., 2018;

Armour et al., 2024), or because the climate system undergoes a profound change under forcing such that the identified past bias does not persist in the future or becomes dwarfed by a larger bias that only emerges in a changing climate (Sanderson et al., 2021). In particular, climate change over recent decades has been characterized by relatively muted radiative feedbacks, likely biasing low constrained estimates of future warming based on observed warming trends (Andrews et al., 2022; Armour et al., 2024).

Here we show that model biases in past OHU may indeed be unable to explain differences in projected OHU, and that previously published constraints were likely influenced by internal climate variability. To narrow the spread in projected OHU we propose a process-based and mechanistically interpretable emergent relationship. This relationship makes it possible to reduce uncertainties in future OHU by accounting for ESM biases in the baseline state of the Southern Hemisphere as quantified by Antarctic summer sea ice extent.

## 2 Methods

### 2.1 Model output

We use output from 28 Earth system models participating in CMIP6 (Table A1) (Eyring et al., 2016). We selected one ensemble member per model based on the availability of necessary output variables in the preindustrial, historical and SSP5-8.5 CMIP6 experiments. When available, output from SSP1-2.6, SSP2-4.5 and SSP3-7.0 scenario experiments was also used. Anomalies relative to the preindustrial state for variables such as heat fluxes, sea ice extent, or thermal expansion were computed by subtracting the matching preindustrial-experiment period from the historical and future variable output starting from the correct experiment branch point. This procedure removes the effect of model drift in the calculated changes (Gupta et al., 2013). The raw preindustrial model output was directly subtracted from the historical and SSP output, without prior processing such as fitting a polynomial regression (Silvy et al., 2022).

OHU is defined as the anomalous net air-sea heat flux (CMIP6 variable `hfds`) integrated in space and cumulatively integrated in time, resulting in units of Joules:

$$\text{OHU}(t) = \int_{t_0}^{t} \int_{\mathcal{A}} \phi(x,y,t') \, \mathrm{d}x \, \mathrm{d}y \, \mathrm{d}t', \tag{1}$$

where $\phi$ is the anomalous net heat flux into the ocean relative to the preindustrial period (units of $\mathrm{W\,m^{-2}}$), $x$ and $y$ are longitude and latitude, $\mathcal{A}$ is the surface area of the ocean, and $t_0 = 1850$ for past OHU and $t_0 = 2024$ for future OHU.

Past and future OHU are defined as OHU over the periods 1850–2023 and 2024–2100, respectively. Since the CMIP6 historical scenario covers 1850–2014 and the SSP scenarios start from 2015, the historical OHU is extended until 2023 using the SSP5-8.5 scenario. We chose SSP5-8.5 because it is the scenario for which most models provide results and because differences across SSP experiments are small over the 2015-2023 period (Riahi et al., 2017).

Antarctic sea ice extent is defined as the total area in which the monthly mean sea ice concentration (CMIP6 variable `siconc`) exceeds 15%.

## 2.2 Estimation of sea level rise due to thermal expansion

As a measure of the direct effect of OHU on sea level, we used the global mean thermal expansion (CMIP6 variable `zostoga`). This variable is available for 20 out of the 28 models. Future global mean thermosteric sea level rise is strongly correlated to future OHU across the model ensemble ($r = 0.97$, $p < 0.05$ two-sided), allowing a direct conversion of OHU to sea level rise based on their ratio of $1.22 \times 10^{-25}\,\mathrm{m\,J^{-1}}$.

## 2.3 Climate feedback parameters

Climate feedback parameters (units: $\mathrm{W\,m^{-2}\,K^{-1}}$) quantify the strength of climate feedbacks that either amplify or dampen the climate system's temperature response to radiative forcing (e.g., Ceppi et al., 2017). Among various feedback components such as surface albedo or lapse rate feedback, the cloud feedback is of particular importance due to its large uncertainty (Zelinka et al., 2020). Cloud feedback arises due to changes in a number of cloud properties including cloud amount, altitude, and optical depth. For the quantification of cloud feedback in this study, we compute spatially resolved climate feedback parameters under the SSP5-8.5 scenario using the radiative kernel method (Soden and Held, 2006) with kernels based on the ERA5 reanalysis (Huang and Huang, 2023). The kernel method consists of systematically applying perturbations in variables of interest (such as temperature, humidity, or albedo) in the radiation code of an atmospheric model and diagnosing the resulting change in shortwave and longwave radiation (Soden et al., 2008).

For each variable $X$ (specifically: temperature, water vapor, and surface albedo), this procedure yields a kernel $K_X$ such that

$$\Delta R_X = K_X \cdot \Delta X, \tag{2}$$

where $R_X$ (in $\mathrm{W\,m^{-2}}$) is the radiative response for variable X with anomaly $\Delta X$ (Huang and Huang, 2023). From this, the climate feedback parameter for variable $X$ can be calculated as $\lambda_X = \Delta R_X / \Delta T$, where $\Delta T$ is the global mean surface temperature anomaly.

The cloud feedback parameter is a special case and can not be directly computed from radiative kernels. Instead, it is computed as a residual of all other terms via

$$\Delta R_{\mathrm{cloud}} = \Delta R - \sum_X \Delta R_X - \mathrm{res}^0, \tag{3}$$

where $\Delta R$ is the total radiative response, and

$$\mathrm{res}^0 = \Delta R^0 - \sum_X \Delta R_X^0 \tag{4}$$

is the residual radiative response under clear sky conditions indicated by the superscript 0 (Huang and Huang, 2023).

## 2.4 Emergent constraint

The posterior probability density functions (PDFs) of ocean heat uptake constrained by sea ice extent observations were calculated using a previously established method (Cox et al., 2013, 2018; Kwiatkowski et al., 2017). Given $N$ realizations of

the response variable $y$ and the predictor variable $x$ as well as their least-squares linear fit $f(x) = a + by$ (in the present case, $N = 28$ climate models provide values for the Antarctic sea ice extent predictor $x$ and the global OHU response $y$), the prediction error is (Cox et al., 2018)

$$\sigma_f(x) = s\sqrt{1 + \frac{1}{N} + \frac{(x - \bar{x})^2}{N\sigma_x^2}}. \tag{5}$$

In the above equation, $s^2$ is the quantity minimized by the linear fit,

$$s^2 = \frac{1}{N-2} \sum_{i=1}^{N} (y_i - f(x_i))^2, \tag{6}$$

while $\bar{x}$ and $\sigma_x^2$ are the ensemble mean and variance of the predictors, respectively. Finally, the constrained PDF $P(y)$ can be calculated as

$$P(y) = \int_{-\infty}^{\infty} P(y|x)P(x)\,\mathrm{d}x, \tag{7}$$

where $P(x)$ is the observational distribution of the predictor, and

$$P(y|x) = \frac{1}{\sqrt{2\pi}\sigma_f(x)} \exp\left(-\frac{(y - f(x))^2}{2\sigma_f(x)^2}\right) \tag{8}$$

is the conditional probability density of $y$ given $x$.

The observational distribution $P(x)$ is assumed to be normal with mean and standard deviation from observations. Where the uncertainty of the observations is not available, an uncertainty is conservatively estimated. For the emergent constraint on future OHU using summer sea ice extent from OSI SAF satellite observations (Fig. 7), we use $\sigma_{\mathrm{obs}} = 1 \times 10^6\,\mathrm{km}^2$, and our results are robust to reasonable changes of this parameter (see section 2.5 below and Fig. A9d for a discussion of this uncertainty).

## 2.5 Observational data

Our principal source of sea ice extent observations for use in the emergent constraint is the OSI SAF Sea Ice index (OSI SAF, 2024) which is based on Advanced Microwave Scanning Radiometer (AMSR) and Special Sensor Microwave Imager/Sounder (SSMIS) instruments, with daily data available starting in 1978. For the sensitivity analysis (Fig. A9), we use two additional satellite microwave radiometry products covering the period 1978–2023 (the NASA Team (DiGirolamo et al., 2022) and Bootstrap (Comiso and Gersten, 2023) products), as well as reconstructions of past sea ice extent from HadISST2.2 (Titchner and Rayner, 2014; Hobbs et al., 2016) and two recent studies (Fogt et al., 2022; Dalaiden et al., 2023).

Interior ocean temperature and salinity were obtained from the World Ocean Atlas 2018 (Garcia et al., 2019), and potential density calculated from these variables using the Gibbs Seawater (GSW) toolbox for Python (McDougall and Barker, 2011).

Ocean heat uptake estimates are from a recent analysis of ocean heat content products (Minière et al., 2023) including an estimate from the international assessment conducted within the Global Climate Observing System (von Schuckmann et al., 2023).

## 2.6 Uncertainty in sea ice extent observations

An estimate of the total uncertainty in daily sea ice concentration due to algorithm and 'smearing' effects from grid interpolation is provided in the gridded OSI SAF sea ice concentration data (OSI SAF, 2017). However, this uncertainty cannot be simply propagated to the calculation of sea ice extent due to spatial and temporal error correlations (Wernecke et al., 2024). An assessment of Arctic sea ice extent uncertainty from a similar satellite observation product has found that the uncertainty in minimum sea ice area in the Arctic is only half of the inter-product spread (Wernecke et al., 2024). Additionally, instrument uncertainties have previously been found to be only $0.036 \times 10^6 \, \mathrm{km}^2$ for Antarctic February sea ice extent in comparable satellite-based sea ice products (Meier and Stewart, 2019).

An alternative approach to gauge the uncertainty of the sea ice extent estimate is to assess the spread of estimates computed from different products. The three satellite-based sea ice concentration products we tested, which use the OSI SAF (OSI SAF, 2017), Bootstrap (Comiso and Gersten, 2023), and NASA Team (DiGirolamo et al., 2022) algorithms, only differ by $0.38 \pm 0.23 \times 10^6 \, \mathrm{km}^2$ in their January-February sea ice extent on average.

Reconstructions of sea ice extent covering decades and centuries preceding the satellite era have larger uncertainties, as illustrated by the spread across the three products (Fig. A10). Nonetheless, there is good agreement between the reconstruction of Fogt et al. (2022) and that of Dalaiden et al. (2023) over the overlapping period, whereas the HadISST2.2 reconstruction shows large, likely spurious step-like variability. We therefore deem the former two reconstructions (Fogt et al., 2022; Dalaiden et al., 2023) to be the most reliable. We use these two reconstructions of annual mean sea ice extent to estimate the range of multi-decadal variability across 40-year periods. We find a maximum difference in sea ice extent between 40-year periods of $0.23 \times 10^6 \, \mathrm{km}^2$ for the period 1850–1980 in the reconstruction of Dalaiden et al. (2023), and a maximum difference of $0.13 \times 10^6 \, \mathrm{km}^2$ for the period 1905–1980 in the reconstruction of Fogt et al. (2022). This is comparable to the CMIP6 multi-model average of historical sea ice extent standard deviation across 40-year periods between 1850–1980 of $0.26 \times 10^6 \, \mathrm{km}^2$. This measure of sea ice multi-decadal internal variability in observations and models is an order of magnitude smaller than the inter-model standard deviation of 1850–1980 mean sea ice extent of $3.3 \times 10^6 \, \mathrm{km}^2$.

In summary, our best estimate of the uncertainty of sea ice extent would be the sum of the uncertainty estimated from the spread between different products ($0.38 \pm 0.23 \times 10^6 \, \mathrm{km}^2$) and the uncertainty that arises from internal variability ($0.23 \times 10^6 \, \mathrm{km}^2$). Here we choose a rather large observational uncertainty of $\sigma_{\mathrm{obs}} = 1 \times 10^6 \, \mathrm{km}^2$ to derive a conservative emergent constraint. Varying this parameter does not change the central constrained estimate but influences the uncertainty reduction (Fig. A9d).

## 2.7 Alternative predictors

The robustness of the constrained result could further be tested by using Southern Ocean cloud cover or deep-ocean temperatures as predictors to constrain OHU, as both are mechanistically linked to Antarctic sea ice extent (Fig. 2). However, a direct comparison between observed and modelled cloud cover requires sampling the CMIP6 ESMs at the same time and location as satellites do. Although this can be done with satellite simulators in ESMs, only 5 out of the 28 ESMs considered here

provide this output. In the case of mean deep-ocean temperature, the limited spatio-temporal density of historical temperature measurements below 2000 meters depth entails that such a predictor would have sizeable uncertainty. Moreover, we find that the relationship between mean deep-ocean temperature and future OHU across the model ensemble ($r = -0.44$, $p < 0.05$) is not as strong and linear as the presently used emergent relationship.

## 3 Results

### 3.1 Antarctic sea ice as an indicator of Southern Hemisphere climate

Antarctic sea ice extent is an indicator of the climate state of the extratropical Southern Hemisphere. Models with greater sea ice extent under preindustrial conditions tend to have colder sea surface temperatures across the Southern Ocean (Fig. 2b) as well as more cloud cover over the mid-latitude Southern Ocean (Kajtar et al., 2021; Shin et al., 2023; Cesana et al., 2025) (Fig. 2a), which modulates radiative heat transfer by reducing downwelling shortwave radiation and enhancing downwelling longwave radiation. Greater sea ice extent is also associated with colder temperatures across the global deep ocean (Fig. 2b), including in deep Atlantic layers mainly ventilated from the North Atlantic (Fig. A1). Given the mean ocean circulation pathways and the long timescales associated with the deep ocean circulation, the plausible causality explaining these correlations is that biases in the temperature of deep ocean waters, much of which ultimately upwell in the high-latitude Southern Ocean, have cascading effects on Southern Hemisphere sea ice, surface temperatures, and clouds (Luo et al., 2023; Sherriff-Tadano et al., 2023).

Under future global warming, ESMs with higher present-day sea ice extent have the potential to lose more sea ice (Kajtar et al., 2021). In particular, under the SSP5-8.5 scenario, many ESMs lose virtually all of their Antarctic summer (January–February) sea ice by 2100, so that summer sea ice loss in 2100 is almost equivalent to baseline sea ice extent (Fig. 3a). Similarly, models with greater preindustrial extratropical and equatorial cloud cover simulate a greater future reduction in cloud cover at these latitudes (Fig. A2), consistent with previous studies (e.g., Thackeray et al., 2024). As a consequence of these links between preindustrial baseline climate and future changes, ESMs with higher preindustrial Antarctic sea ice extent tend to experience a greater shift in their simulated Southern Hemisphere climate in the future. This shift in climate manifests itself through greater warming of the surface atmosphere and ocean in the Southern Hemisphere (Fig. 3b,c), a more positive global cloud feedback (Fig. 3d), and consequently greater global OHU (Fig. 7). This additional OHU in models with higher preindustrial Antarctic sea ice extent is particularly pronounced in the Southern Hemisphere mode and intermediate waters (Fig. 4) which tend to transport heat northwards and into the interior ocean (Armour et al., 2016).

The cloud feedback links preindustrial Antarctic sea ice extent and future global OHU. Across the ESM ensemble, this connection is globally apparent by the end of the 21st century as strong correlations between cloud feedback and Antarctic sea ice extent loss (Fig. A3a), and between cloud feedback and global OHU (Fig. A3b). The global extent of this relationship between Antarctic sea ice extent and extent loss, cloud feedback and OHU is the result of a northward propagation of this relationship originating in the Southern Ocean. The surface warming signal in the ocean and atmosphere related to preindustrial sea ice extent first emerges in the southern high latitudes around 1990–2010, gradually spreading northwards and covering most of the Southern Hemisphere by 2030–2050 under SSP5-8.5 (Fig. 5). This causes a concomitant spreading of sea ice–related

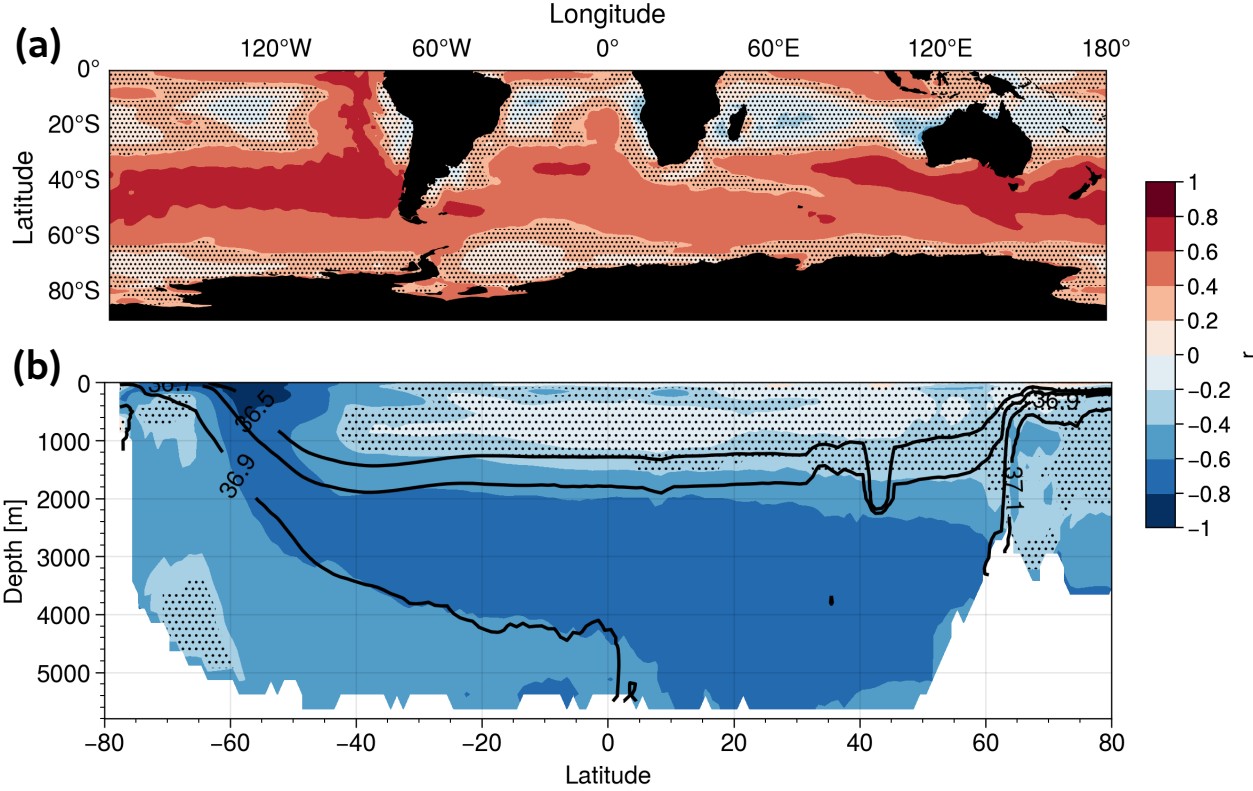

**Figure 2. Atmospheric and oceanic connections to Antarctic sea ice extent in the preindustrial state. (a)**, Inter-model correlation between preindustrial annual-mean Antarctic sea ice extent and preindustrial total cloud cover in the Southern Hemisphere. In red areas, local cloud cover is increased for models with higher sea ice extent. **b**, Inter-model correlation between preindustrial annual-mean Antarctic sea ice extent and preindustrial zonal mean ocean temperature across all ocean basins. In blue areas, local seawater is colder for models with higher sea ice extent. Black contours show zonal mean potential density relative to a reference pressure of $2000\,\mathrm{dbar}$ from observations (Garcia et al., 2019). In both panels, stippling indicates regions where the correlation is not significant ($p \geq 0.05$, two-sided).

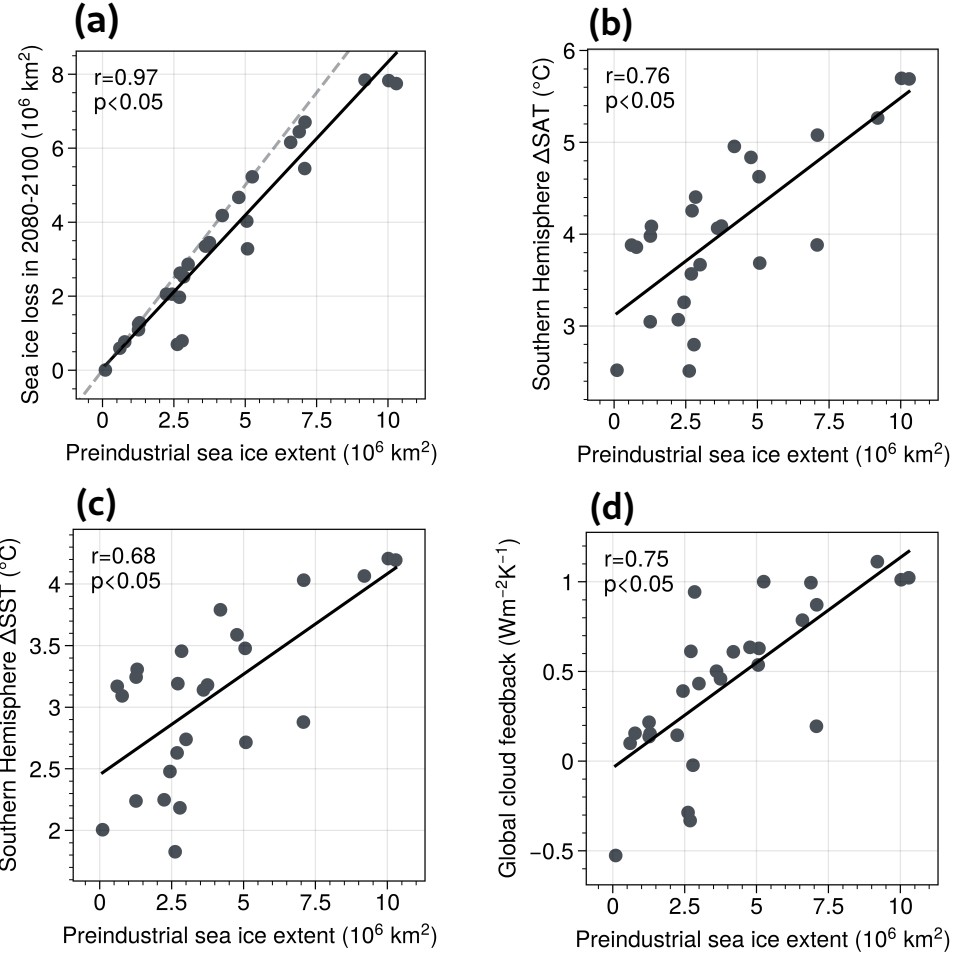

**Figure 3. Links between preindustrial Antarctic sea ice and Southern Hemisphere climate change.** CMIP6 inter-model relationship between preindustrial Antarctic summer (January–February) sea ice extent and future sea ice extent loss **(a)**, Southern Hemisphere surface air temperature increase **(b)**, Southern Hemisphere sea surface temperature increase **(c)**, and global mean cloud feedback parameter **(d)**. In each panel, the black line shows the least squares linear regression fit, and the Pearson correlation coefficient $r$ and two-sided $p$-value are given in the upper left corner. The y-axis of all panels represents anomalies between years 2080–2100 of the high-emissions SSP5-8.5 scenario and the preindustrial state.

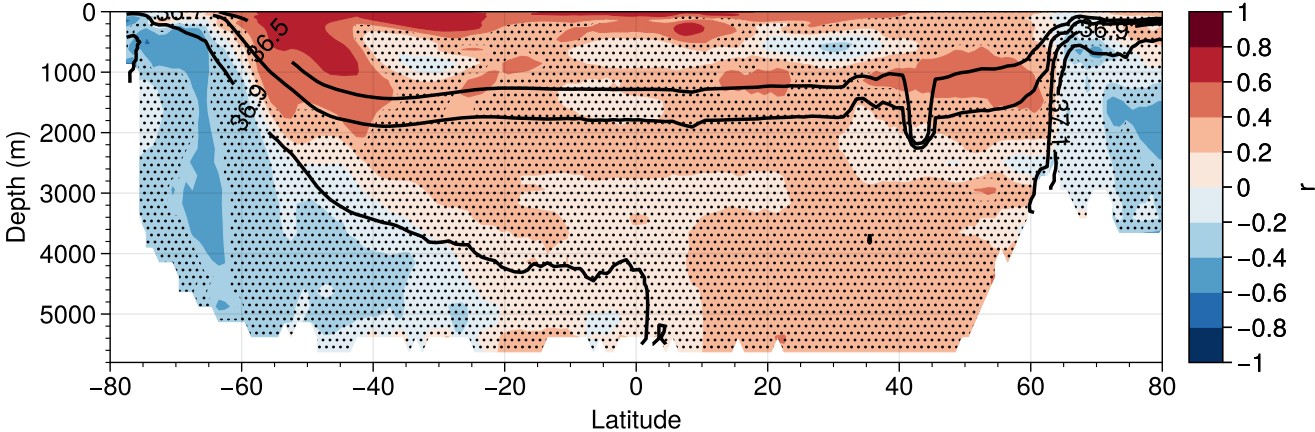

**Figure 4. Zonal mean ocean warming related to preindustrial sea ice extent.** Correlation coefficient across the ensemble of CMIP6 models between preindustrial annual mean Antarctic sea ice extent and zonal mean ocean warming in 2080–2100 under SSP5-8.5 relative to preindustrial. Red shading indicates regions where models with more preindustrial sea ice tend to have more ocean warming in the future scenario. Stippling indicates regions where the correlation is not significant ($p \geq 0.05$, two-sided). Black contours show zonal mean potential density referenced to 2000 dbars from observations (Garcia et al., 2019).

local cloud feedback starting from the Southern Ocean and attaining its near-global extent by mid-century (Fig. A4). Although cloud feedback is in general controlled by a number of contributions including cloud amount, altitude, and optical depth (Zelinka et al., 2016; Ceppi et al., 2017), the signal is apparent in total cloud amount (Fig. A2). The northward propagation of these significant inter-model relationships likely results from anomalous heat transport in the ocean and/or the atmosphere (England et al., 2020b, a; Ayres et al., 2022; Luo et al., 2025) resulting in a teleconnection from the Southern Ocean to the tropical oceans via midlatitude cloud feedback (e.g., Zhang et al., 2021; Zhang and Deser, 2024; Ford et al., 2025).

Decomposing cloud feedback into its shortwave and longwave radiative components reveals that the global relationship between sea ice loss and cloud feedback is mostly mediated by the shortwave component (Fig. A3c–d), whereas the longwave component remains restricted to the Southern Ocean by the end of the 21st century. Furthermore, partitioning the excess OHU into its individual air-sea heat flux components demonstrates that the higher OHU in models with greater Antarctic sea ice loss is mainly due to increased shortwave-driven OHU in the Southern Hemisphere and globally increased sensible OHU (Fig. A5). The increased sensible OHU is a direct consequence of the stronger atmospheric warming in models with more Antarctic sea ice loss. The increased shortwave-induced and sensible OHU associated with larger Antarctic sea ice loss is slightly counteracted by a reduced latent air-sea heat flux at low latitudes (Fig. A5h). As sea ice loss strongly accelerates after 2024, these relationships emerge only for future (2024–2100) OHU and are not apparent for OHU over the historical (1850–2024) period.

We emphasize that changes in summer Antarctic sea ice extent are likely not the primary cause of global cloud and temperature changes. Rather, Antarctic sea ice extent is an indicator and integral part of the baseline state of the extratropical Southern

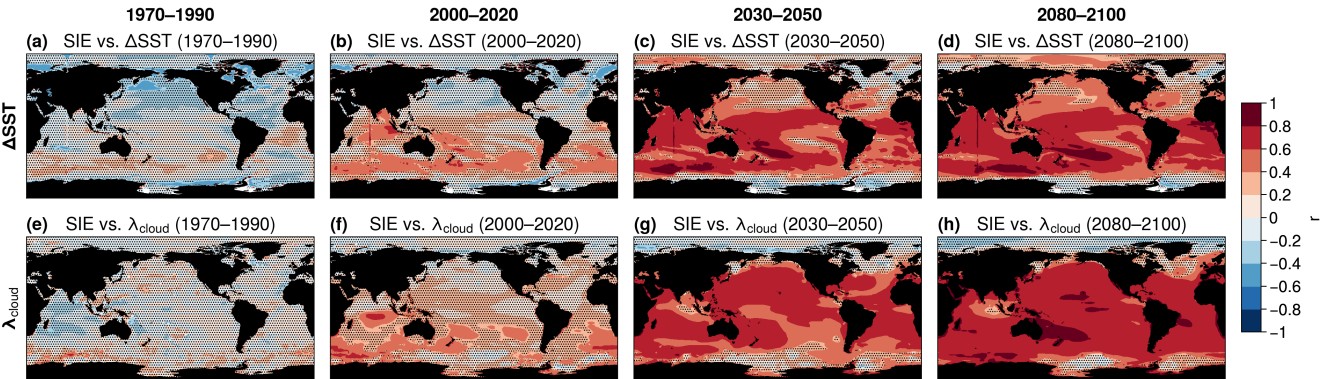

**Figure 5. Time evolution of sea ice–related surface warming and cloud feedback.** Inter-model correlation across CMIP6 models under SSP5-8.5 between preindustrial Antarctic summer sea ice extent and (top row) local sea surface temperature anomaly, (bottom row) local total cloud feedback parameter $\lambda_{cloud}$, during different 20-year periods between 1970 and 2100. In all panels, stippling indicates regions where the correlation is not significant ($p \geq 0.05$, two-sided).

Hemisphere climate (Fig. 2; Kajtar et al., 2021; Luo et al., 2025; Ford et al., 2025), which itself preconditions projected global climate warming. This idea is schematically illustrated in Fig. 6 which shows how the amplitude of climate warming is pre-conditioned by the initial climate state. The summer extent of Antarctic sea ice can thus be regarded as measuring a potential for future change (Fig. 3). Nonetheless, the loss of Antarctic sea ice does have direct local influences (Kay et al., 2014). Any reduction of white sea ice cover exposes the underlying ocean, allowing more heat to be absorbed. While the additional OHU under the previously covered sea ice is small compared to the global OHU (about 6% in the multi-model mean), this additional warming close to the sea ice edge further accelerates the loss of sea ice cover through surface albedo feedback and contributes to the link between present-day sea ice and future climate change (Stolpe et al., 2019).

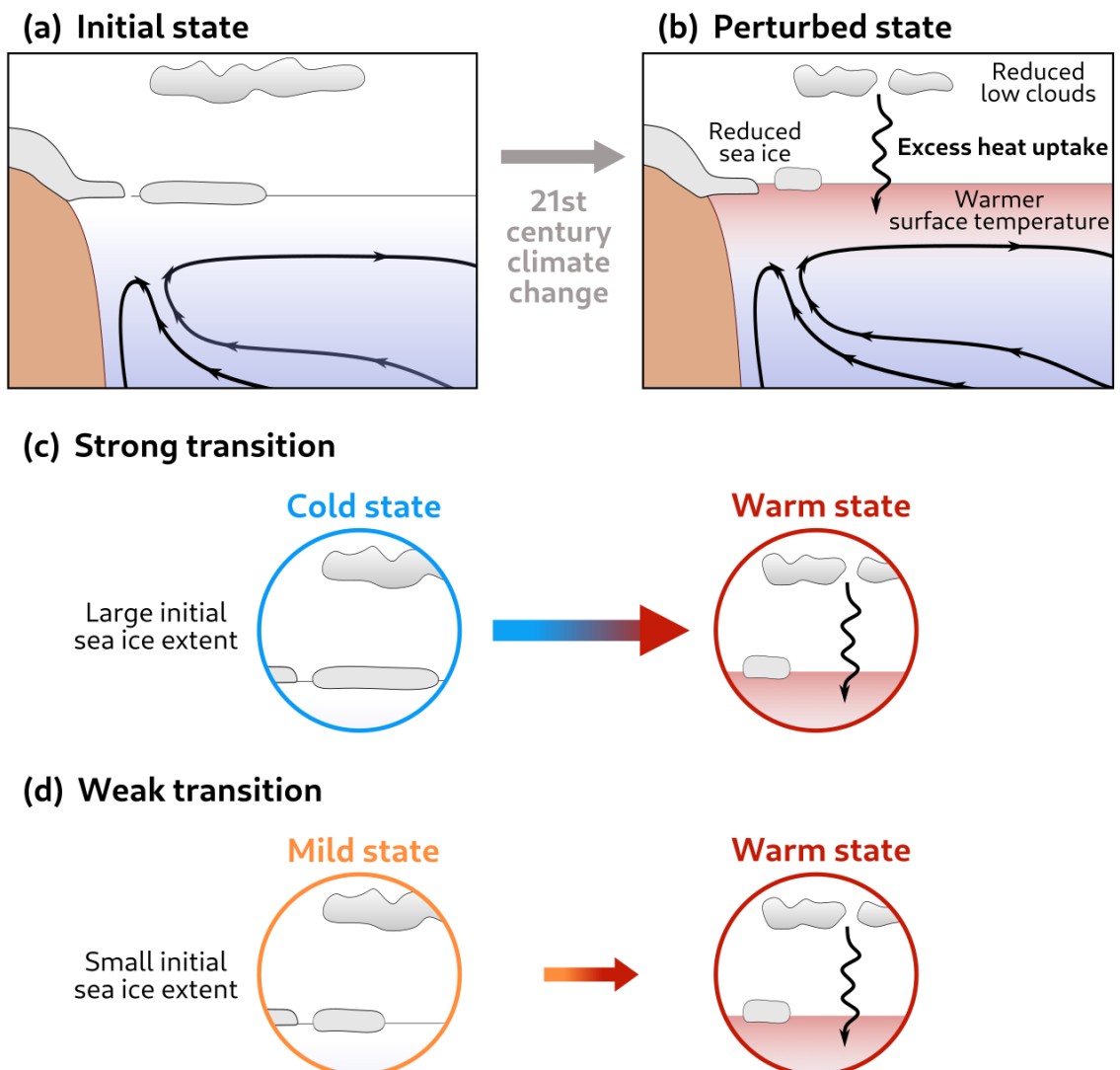

**Figure 6. Schematic representation of the link between historical Antarctic sea ice extent and future ocean heat uptake.** Under 21st century climate change, the Southern Hemisphere climate system transitions from an initial state **(a)** to a perturbed state **(b)** characterized by reduced sea ice, surface ocean and atmospheric warming, and reduced lower cloud cover. Crucially, the amplitude of this transition in each climate model, and therefore the magnitude of future ocean heat uptake, depends on the model's initial climate state: a cold initial state characterized by large sea ice extent leads to a strong transition with high heat uptake **(c)**, while a mild initial state characterized by small sea ice extent leads to a weaker transition with less heat uptake **(d)**.

## 3.2 Emergent constraints on future change

The mechanistic understanding and inter-model relationships presented show that model bias in baseline sea ice extent in austral summer is a physical indicator of future sea ice loss, surface warming, and cloud feedback (Fig. 3). As cloud feedback mediates future OHU (Fig. A3), historical observations of Antarctic sea ice can be used to constrain future OHU (Fig. 7). Using the 1980–2020 summer sea ice extent from the OSI SAF satellite observational product (OSI SAF, 2017) of $4.41 \pm 1.00 \times 10^6 \, \mathrm{km}^2$ to constrain future OHU results in an estimate of future global OHU between 2024–2100 of $1244 \pm 141 \, \mathrm{ZJ}$ under SSP1-2.6 (Fig. 7a-b) and $2595 \pm 209 \, \mathrm{ZJ}$ under SSP5-8.5 (Fig. 7c-d, results for SSP2-4.5 and SSP3-7.0 are shown in Fig. A6 and detailed in Table A2). The constrained median estimate is 3% higher and 14% less uncertain than the CMIP6 ensemble prior median under SSP1-2.6, and 14% higher and 33% less uncertain under SSP5-8.5.

In all four SSPs considered, the correlation between 1980–2020 sea ice extent and future OHU is above 0.6 and statistically significant at the $p < 0.05$ level according to a two-sided Student's $t$-test (Table A2). This suggests that the identified relationships are robust and explain a substantial fraction of inter-model spread in future OHU irrespective of the scenario. Given our conservative choice of predictor uncertainty and available model ensemble sizes (Methods), the difference between unconstrained and constrained OHU mean values is statistically significant under SSP2-4.5 and SSP5-8.5 but not under SSP1-2.6 ($p = 0.11$) and SSP3-7.0 ($p = 0.09$) according to a two-sided two-sample Student's $t$-test (Table A2).

The higher OHU estimates directly translate to greater than currently anticipated future sea level rise due to thermal expansion. Under SSP1-2.6 the constrained thermosteric global mean sea level rise from 2024 to 2100 is $15.2 \pm 1.7 \, \mathrm{cm}$, assuming a constant conversion factor between OHU and thermosteric sea level rise (see Methods). Under SSP5-8.5, the constrained estimate is $31.6 \pm 2.5 \, \mathrm{cm}$. Both estimates are higher and less uncertain than the respective unconstrained estimates of $14.5 \pm 2.0 \, \mathrm{cm}$ and $29.5 \pm 3.8 \, \mathrm{cm}$ (Table A2).

The relationships uncovered here can also help to constrain the strength of cloud feedback and the magnitude of global warming by the end of the 21st century. Present-day Antarctic sea ice extent is significantly correlated with future cloud feedback and global warming in all four SSPs considered (Table A2). Using 1980–2020 summer sea ice extent as a predictor, global mean cloud feedback is constrained to be 19% and 31% higher than the CMIP6 median under SSP1-2.6 and SSP5-8.5, respectively (Fig. 8b). Future global mean surface air warming is constrained to be 3–7% greater than the CMIP6 median (Fig. 8c). The uncertainty in the estimates is reduced by 18% for cloud feedback and by 11% for surface warming under SSP5-8.5 (results for other SSPs are shown in Fig. A7 and detailed in Table A2). The uncertainty reduction for warming and cloud feedback is smaller than for OHU because present-day sea ice extent is more strongly correlated with future OHU ($r = 0.87$ under SSP5-8.5) than with end-of-century cloud feedback ($r = 0.71$) or surface air warming ($r = 0.61$).

The tighter constraint on OHU may be explained by two factors. First, the correlation between Antarctic sea ice extent and local cloud feedback is particularly strong over the southern mid-latitudes where OHU is most efficient (Armour et al., 2016) (Fig. A3) and where much of the additional OHU occurs (Fig. 4). Second, larger baseline Antarctic sea ice is associated with colder deep waters (Fig. 2b), whose exposure to the warming atmosphere in the Southern Ocean can promote OHU through sensible heat flux (Fig. A5).

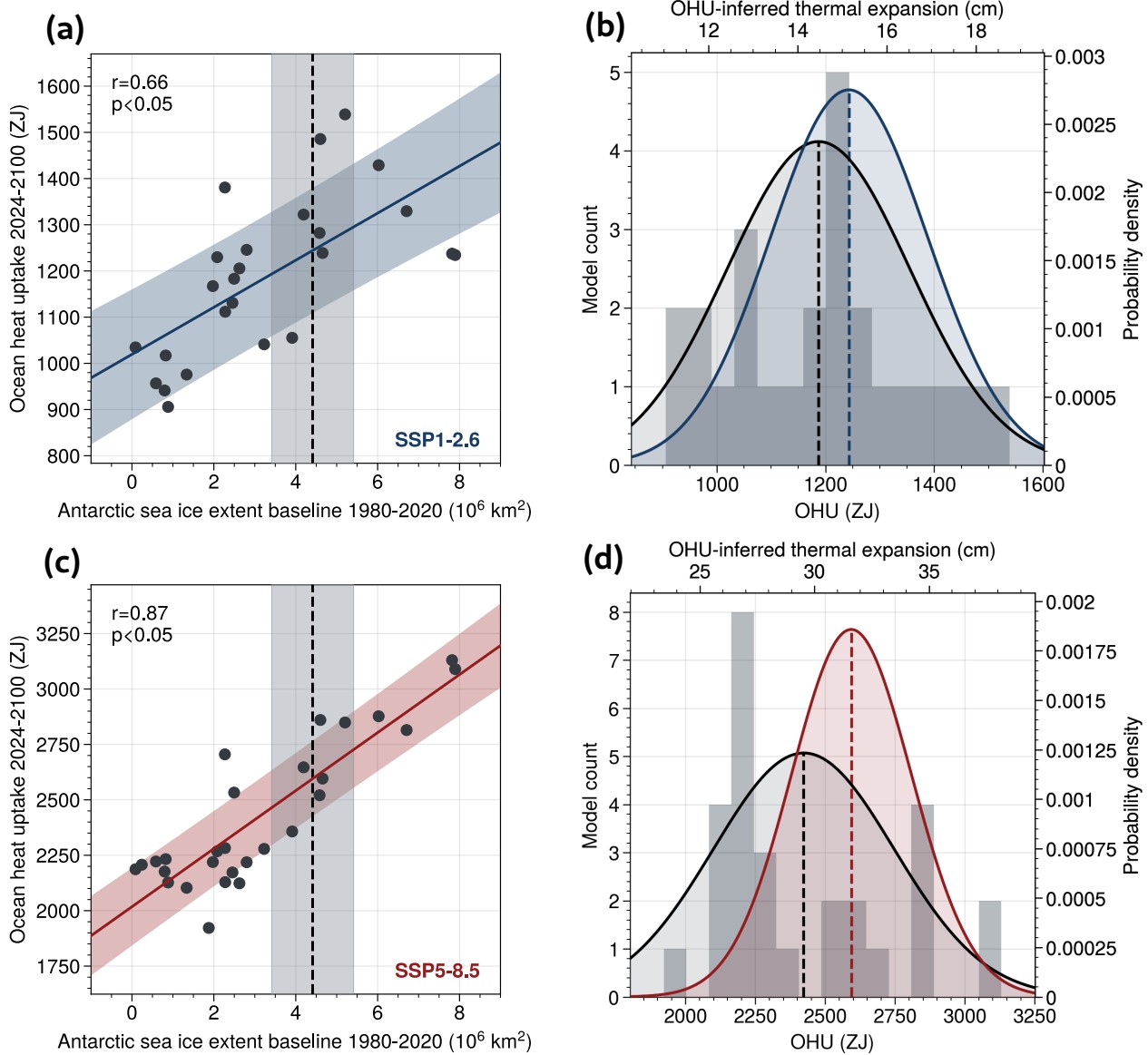

**Figure 7. Emergent constraint on future global ocean heat uptake. a**, Inter-model relationship between 1980-2020 Antarctic summer (January–February) sea ice extent and cumulative global OHU over 2024–2100 under the SSP1-2.6 scenario. The blue line and shading show the least squares linear regression fit and its uncertainty (see Methods), with the Pearson's correlation coefficient $r$ and two-sided $p$-value given in the upper left corner. The dashed vertical line shows satellite observations of Antarctic summer sea ice extent averaged over 1980–2020 (OSI SAF, 2017) and the grey shading shows the associated uncertainty of $1 \times 10^6 \, \text{km}^2$; this relatively large observational uncertainty ensures we derive a conservative emergent constraint (Methods). **b**, Unconstrained prior (black) and constrained posterior (blue) probability density functions of 2024–2100 global OHU. In grey we show the prior histogram for 2024–2100 OHU (Methods). **c**, as panel **a** but for the SSP5-8.5 scenario. **d**, as panel **b** but for the SSP5-8.5 scenario.

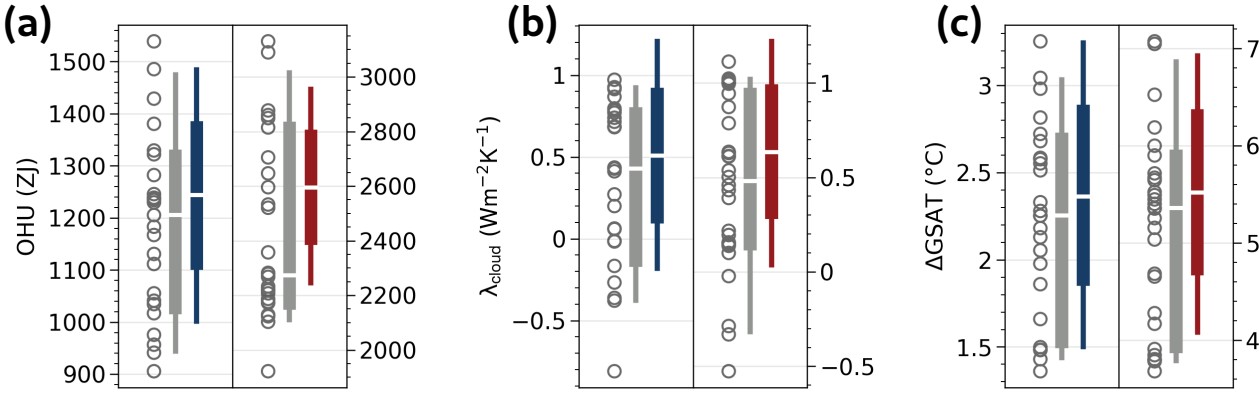

**Figure 8. Constrained distributions of global OHU, cloud feedback, and warming.** Prior and constrained distributions of **(a)** cumulative global OHU from 2024 to 2100, **(b)** global mean cloud feedback parameter in 2080–2100, and **(c)** global mean surface air temperature (GSAT) anomaly in 2080–2100 relative to the preindustrial. In each panel, distributions are shown for SSP1-2.6 (left) and SSP5-8.5 (right). The grey circles and grey boxplots show the prior distribution of model values, and the blue and red boxplots show the constrained distributions for SSP1-2.6 and SSP5-8.5, respectively. In each boxplot, the white line shows the median, the central box spans the likely range (66%), and the whiskers extend to a 95% confidence interval. The constrained values are normally distributed by construction (Methods). Note that the $y$-axis scale is different between the two SSPs in panels **a** and **c**.

For completeness, we also test whether sea ice extent can be used to constrain past (1850–2024) OHU. We find no significant emergent relationship between baseline Antarctic sea ice and historical OHU. The inter-model correlation coefficient between January-February Antarctic sea ice extent and 1850–2024 OHU is $r = -0.03$ for preindustrial mean sea ice extent and $r = -0.04$ for 1980–2020 mean sea ice extent. The non-existing correlation over the past indicates that the sea-ice linked feedback we identify has not yet influenced past OHU, warming, or cloud feedback, although it will affect their future.

To facilitate comparison with previous studies which used past warming trends as predictors to constrain future OHU (Lyu et al., 2021) or global surface warming (Tokarska et al., 2020), we now apply our emergent constraint to the same uncertain variables considered in these two studies. For future 0–2000m OHU under SSP5-8.5 in 2081–2100 relative to 2005–2019 as in Lyu et al. (2021), we obtain a constrained estimate which is 16% (9%) higher than the unconstrained CMIP6 median (mean), in contrast to Lyu et al. (2021) whose constrained OHU estimate was 10% lower than the prior mean (Fig. A8a–b). Historical Antarctic sea ice extent provides higher predictive skill for future 0–2000m OHU ($r = 0.9$) than does past 0–2000m OHU ($r = 0.72$ in Lyu et al. (2021)). For future global surface air temperature warming under SSP5-8.5 in 2081–2100 relative to 1850–1900 as in Tokarska et al. (2020), we obtain a constrained estimate which is 5% (7%) higher than the unconstrained CMIP6 median (mean), again in contrast to the constrained estimate from Tokarska et al. (2020) which was 14 % lower than the prior mean (Fig. A8c–d).

### 3.3 Robustness of the emergent constraint

In these constrained projections we used the satellite-observed summer (January-February) sea ice extent averaged over 1980-2020 as the observable climate variable. Similar results are obtained when alternative definitions of the observable variable are employed (Fig. A9). Different satellite observational products lead to very minor shifts in the constrained OHU projection, indicating that observational uncertainty in present-day sea ice extent is sufficiently small (much less than the specified uncertainty of $1 \times 10^6 \, \mathrm{km}^2$ in Fig. 7) to obtain robust uncertainty reduction (Fig. A9a,d and Methods). Using annual mean or austral winter sea ice extent or different definitions of the summer season also yield broadly consistent uncertainty reductions (from $-13\%$ to $-38\%$) and OHU increases (from $+3\%$ to $+11\%$) under SSP5-8.5 (Fig. A9c).

Antarctic sea ice cover shows both inter-annual and multi-decadal variability over the satellite record (Fig. A10), so that the choice of baseline period can affect our emergent constraint. Choosing 1980-2000, 1990-2010 or 2000-2020 instead of 1980-2020 as baseline periods within the satellite record yields constrained OHU estimates of $+5\%$, $+8\%$ or $+9\%$ above the unconstrained mean, respectively. This relatively small sensitivity stems from the large inter-model spread in Antarctic sea ice extent across CMIP6 models compared to observed variability since 1980 (Fig. A10). Reconstructions of Antarctic sea ice cover over earlier parts of the 20th century and preceding centuries possess larger uncertainties (Dalaiden et al., 2023; Fogt et al., 2022; Titchner and Rayner, 2014), yet they also indicate a negative bias of the multi-model mean annually averaged extent (Fig. A10). Consequently, choosing different 40-year baseline periods between 1920 and 2000 in these reconstructions (Methods) leads to a constrained heat uptake between 3–12% higher than the CMIP6 mean under the SSP5-8.5 scenario (Fig. A9b).

For further robustness testing, we examine the correlation between historical sea ice extent and future OHU (Fig. 7a,c) in an out-of-sample test using 16 models from the CMIP5 ensemble, and we probe the sensitivity of this correlation to the chosen OHU time period and sea ice seasonality in both CMIP5 and CMIP6 ensembles (Fig. A11). In the CMIP6 ensemble, maximal correlation between historical sea ice extent and future OHU under SSP5-8.5 is obtained for summer sea ice extent together with an OHU time period starting at any year after 1850 and ending after approximately 2070 (Fig. A11a–b). For time periods ending prior to 2030 the correlation becomes statistically insignificant, underlining the fact that the mechanism underlying the emergent relationship occurs only under future forcing. In the CMIP5 ensemble, correlations are higher for annual mean sea ice extent, but the temporal structure is similar to CMIP6 with maximal correlations for OHU periods extending towards the end of the 21st century (Fig. A11c–d).

The correlation between historical sea ice extent and future OHU is not an artifact of outliers or caused by individual model values of sea ice extent or OHU far from the center of the multi-model distribution (Fig. A12). A significant positive correlation persists across all considered SSPs even when discarding several models with the highest or lowest values of sea ice extent (Fig. A12a,c) and OHU (Fig. A12b,d). Furthermore, using a Huber loss function instead of ordinary least squares (OLS) in order to reduce the influence of outliers yields an almost identical regression slope ($131 \times 10^{-6} \, \mathrm{ZJ/km}^2$ for OLS, $130 \times 10^{-6} \, \mathrm{ZJ/km}^2$ for Huber) and coefficient of determination ($r^2 = 0.75$ for both methods under SSP5-8.5).

The robustness of the constrained results can further be corroborated by observations of cloud cover and deep-ocean temperatures. Though these observations are not readily used as formal predictors in an emergent constraint (see Methods), they show that ensemble mean simulated global deep-ocean temperatures are 7% higher than observations and that simulated mid-latitude Southern Ocean (30–50°S) cloud cover is 7% less than in satellite observations. The underestimation of cloud cover and overestimation of deep ocean temperatures in ESMs concur with a negative bias in Antarctic sea ice extent (Fig. 2), and with underestimated cloud feedback, atmospheric warming, and OHU over the 21st century in the unconstrained CMIP6 ensemble mean (Figs. 3 and 8).

## 4   Conclusions

The increased estimates of OHU, cloud feedback, and global warming found here are consistent with increased low-cloud feedback estimates by recent observational constraints (Ceppi et al., 2024; Wu et al., 2025; Aerenson and Marchand, 2025), but contrast with previous studies that suggest an overestimation of the future warming by CMIP6 ESMs based on past warming and OHU trends (Tokarska et al., 2020; Lyu et al., 2021; Nijsse et al., 2020; Jiménez-de-la Cuesta and Mauritsen, 2019). A possible explanation for this difference is the limited length and representativeness of the observational records from 1980 to 2015 employed in these studies for the underlying long-term climate change (Andrews et al., 2022; Armour et al., 2024). The 1980-2015 period has been marked by patterns of sea surface temperature change associated with weaker climate feedbacks than expected under long-term climate change (Andrews et al., 2022). These patterns, which include surface cooling in the eastern tropical Pacific and parts of the Southern Ocean, are less likely than 5% across CMIP5 and CMIP6 simulations (Wills et al., 2022). This mismatch between models and observations can bias emergent constraints that use trends over the 1980-2015 period (Andrews et al., 2022; Armour et al., 2024). More generally, climate variability is a critical confounding factor when short-length observational records are employed to constrain projections. As an example, shifting the 2005–2019 observational period for past OHU trends used in Lyu et al. (2021) only six years earlier (1999–2013) results in statistically insignificant relationship between past OHU trend and future OHU (Fig. A13).

Similarly, satellite observations of Antarctic sea ice could coincide with a period of anomalously large or small sea ice extent, biasing our emergent constraint. To test our results for such potential bias, we used different baseline periods for sea ice extent, including periods before the satellite era for which reconstructions of Antarctic sea ice are available (Dalaiden et al., 2023; Fogt et al., 2022; Titchner and Rayner, 2014). We find that our mechanism-based emergent constraint consistently reduces uncertainty and increases OHU projections, even with the substantial uncertainty we attribute to the predictor (Methods). This robustness of our constraint stems from the use of an observable mean-state variable—instead of observable trends, which tend to be more sensitive to transient (decadal) anomalies—and from the strength of the emergent relationship of Fig. 7c ($r = 0.87$).

Another potential factor for the difference between the present and previous estimates of OHU and atmospheric warming (Tokarska et al., 2020; Lyu et al., 2021; Nijsse et al., 2020; Jiménez-de-la Cuesta and Mauritsen, 2019) is the inability of past trends to account for a future regime shift in the climate system (Marvel et al., 2018; Armour et al., 2024; Liang et al., 2024). The climatic relationships and feedbacks underpinning our emergent constraint are dependent on a shift in the Southern Hemi-

sphere climate state under pronounced greenhouse forcing (Fig. 6), exemplified by the near-total disappearance of Antarctic summer sea ice under a high-emissions scenario (Fig. 3a). Indeed, the constraint is stronger for higher emissions scenarios (Fig. 7), and is invalid for past OHU, indicating that the processes presented here dominate inter-model spread only under moderate to strong forcing. Similarly, the OHU constraint based on past warming trends (Lyu et al., 2021) is insignificant for initial time periods ending before 2010 but becomes stronger for time periods chosen later in the 21st century (Fig. A13), which suggests that the potential regime shift connected to cloud feedback (Fig. 5) is necessary for obtaining a strong constraint. Although Antarctic sea ice extent has long seemed relatively unresponsive to anthropogenic forcing, the recently observed abrupt sea ice loss in 2016 and the historical minimum extent anomaly in 2023 have highlighted the possibility of an ongoing regime shift (Hobbs et al., 2024). These observed sea-ice changes could foreshadow stronger Southern Hemisphere climate feedbacks and ocean warming in coming decades (Kang et al., 2023).

Our results suggest more warming and heat uptake than the CMIP6 multi-model mean, in contrast to previous studies (Tokarska et al., 2020; Lyu et al., 2021; Nijsse et al., 2020). Our results thus confirm the recent finding that these studies may have underestimated future warming and that the very low equilibrium climate sensitivity (ECS) estimates of some climate models are unlikely (Myhre et al., 2025). Nevertheless, our results do not invalidate previous results indicating that that the extreme end of strong warming and cloud feedback of high-ECS CMIP6 projections is unlikely (Tokarska et al., 2020; Lyu et al., 2021; Nijsse et al., 2020; Jiménez-de-la Cuesta and Mauritsen, 2019; Myers et al., 2021; Cesana and Del Genio, 2021). Furthermore, other shared biases in CMIP6 models could potentially imply additional positive or negative corrections to future OHU projections (e.g., Wang et al., 2024). Ideally, if other such biases are identified in the future, they could be combined with our findings to arrive at a combined observational constraint (e.g., Bretherton and Caldwell, 2020; Terhaar et al., 2022). Endeavours to identify and correct such biases thus remain of utmost importance.

The relationships between oceanic, cryospheric and atmospheric variables revealed in this study provide guidance for the reduction of important mean state biases in ESMs. Specifically, they highlight the need for an accurate representation of clouds in ESMs, as well as the importance of reducing uncomfortably large biases in the deep ocean hydrography. Similar to clouds, deep ocean temperatures explain an important part of differences in present-day Antarctic sea ice and clouds (Fig. 2) and thereby influence the future climate change in CMIP6 models (Fig. 6). Improving ocean circulation and hydrography for climate projections therefore requires additional attention (Luo et al., 2023; Sherriff-Tadano et al., 2023), alongside efforts to improve the simulation of clouds (Hyder et al., 2018; Zelinka et al., 2020).

Overall, our results imply that potent feedback mechanisms at mid to high southern latitudes may cause future ocean heat uptake to be higher than expected from previous assessments. Increased ocean heat uptake would cause more thermosteric sea level rise (our central estimates for total end-of-century sea level rise are between 15–31 cm, depending on scenario), more damage to marine ecosystems and create additional risks to socio-economic systems. Moreover, increased cloud feedback and consequently larger future warming will make it even harder to limit warming in order to reach climate targets, for example those set by the Paris Agreement. This prospect calls for improved projections of coupled ocean-atmosphere climate feedbacks, continued monitoring of variability and trends across the Southern Ocean, as well as imminent and strong reductions in greenhouse gas emissions.

*Data availability.* Preprocessed time series of Antarctic sea ice extent, ocean heat uptake, global mean sea surface temperature and atmospheric temperature used in this study are available at: https://doi.org/10.5281/zenodo.15693980.

Observational and model data used in this study are available at the following locations:

- CMIP6 model output: https://esgf-node.llnl.gov/projects/cmip6/

- World Ocean Atlas ocean temperature and salinity data: https://www.ncei.noaa.gov/archive/accession/NCEI-WOA18

- Radiative kernels from Huang and Huang (2023): https://doi.org/10.17632/vmg3s67568

- Ocean heat content time series from Minière et al. (2023): included in this study's Zenodo repository (https://doi.org/10.5281/zenodo.15693980).

- Cloud cover observational data: https://doi.org/10.24381/cds.68653055

- OSI SAF sea ice extent data: https://doi.org/10.24381/cds.3cd8b812

- HadISST2.2 sea ice extent data: https://www.metoffice.gov.uk/hadobs/hadisst2/data/download.html

- Sea ice extent reconstruction from  Fogt et al. (2022): https://doi.org/10.6084/m9.figshare.c.5709767.v1.

- Bootstrap algorithm sea ice extent data: https://nsidc.org/data/nsidc-0079/versions/4#anchor-1.

- NASA Team algorithm sea ice extent data: https://climatedataguide.ucar.edu/climate-data/sea-ice-concentration-data-nasa-goddard-

**Appendix A**

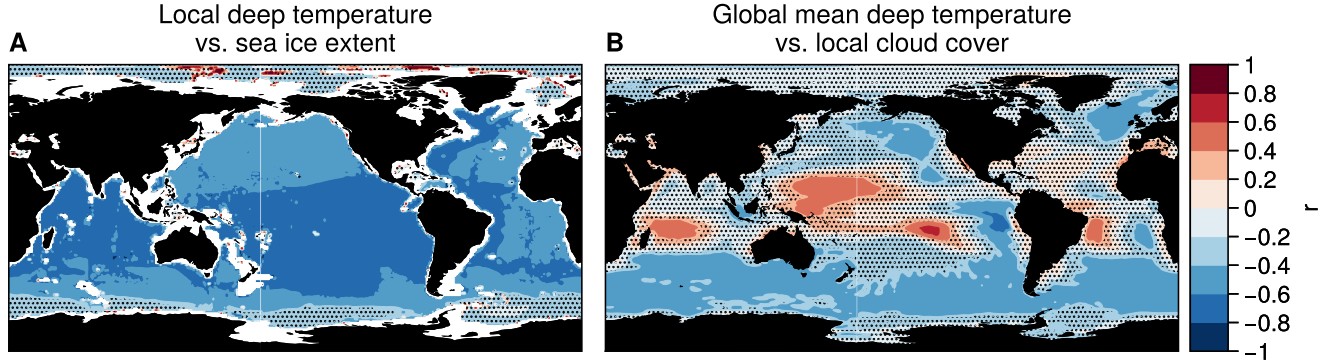

**Figure A1. Relationship between deep ocean temperature and preindustrial surface climate. a**, Inter-model correlation between preindustrial local deep ocean temperature (averaged over 2000-4000 m depth) and preindustrial Antarctic annual mean sea ice extent. **b**, Inter-model correlation between preindustrial global mean deep ocean temperature (averaged over 2000-4000 m depth) and preindustrial local total cloud cover. In both panels, stippling indicates regions where the correlation is not significant ($p \geq 0.05$, two-sided).

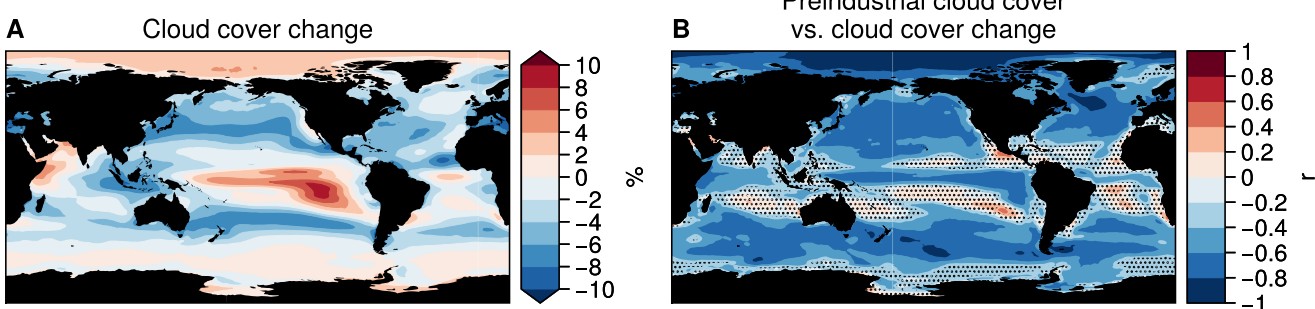

**Figure A2. Changes in cloud cover. a,** Change in total cloud cover in 2080-2100 under SSP5-8.5 relative to preindustrial. **b,** Inter-model correlation between local preindustrial cloud cover and local cloud cover change. Blue regions indicate that models with high local initial cloud cover lose more local cloud cover. In panel **a**, the unit of % is the unit of total cloud cover and does not refer to a relative change. In panel **b**, stippling indicates regions where the correlation is not significant ($p \geq 0.05$, two-sided).

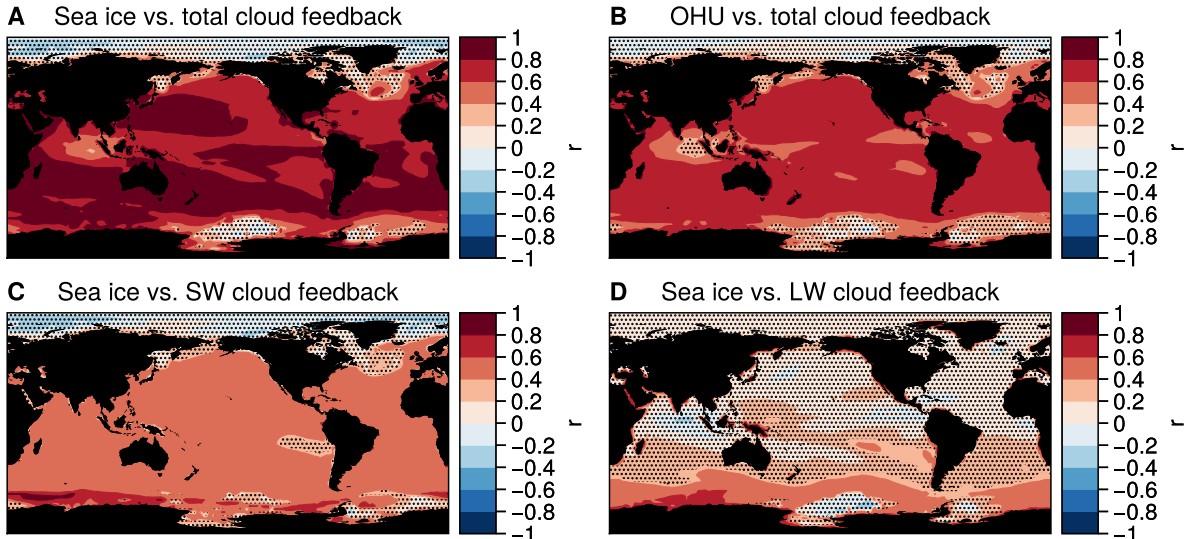

**Figure A3. Relationship between the local cloud feedback and anomalies in sea ice extent and OHU.** Inter-model correlation across CMIP6 models under SSP5-8.5 between **a**, local net cloud feedback parameter and Antarctic summer sea ice extent loss by 2080-2100; **b**, local net cloud feedback parameter and total ocean heat uptake from 2024–2100; **c**, as for **a** but with shortwave cloud feedback parameter; and **d**, as for **a** but with longwave cloud feedback parameter. Stippling indicates regions where the correlation is not significant ($p \geq 0.05$, two-sided). In panels **a**, **c** and **d**, red areas indicate locations where models with greater Antarctic sea ice loss tend to have more positive local cloud feedback. In panel **b**, red areas indicate locations where models with more positive local cloud feedback tend to have greater global 2024–2100 OHU.

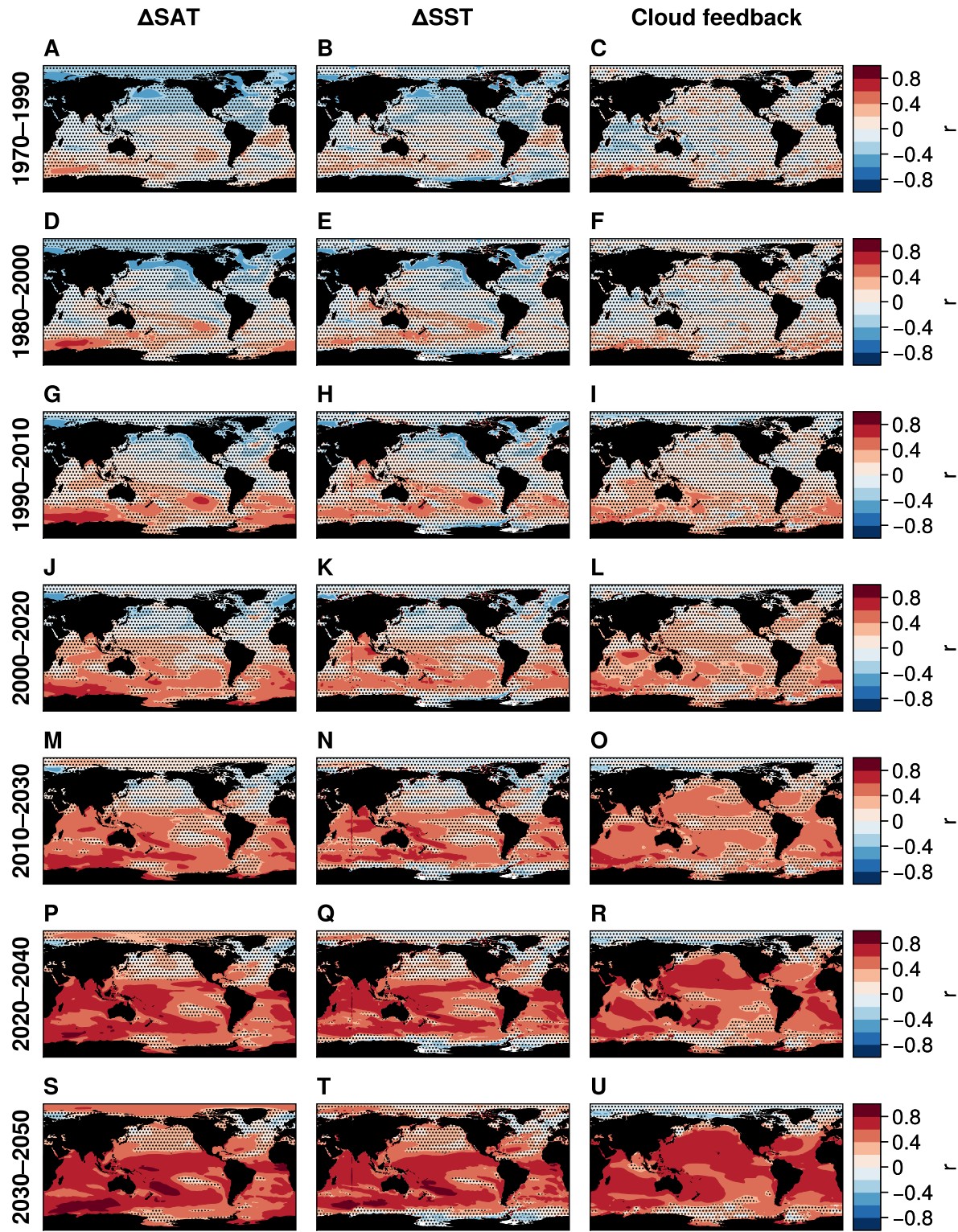

**Figure A4. Time evolution of sea ice–related climate change.** Inter-model correlation across CMIP6 models under SSP5-8.5 between preindustrial Antarctic summer sea ice extent (SIE) and (left column) local surface air temperature anomaly, (middle column) local sea surface temperature anomaly, and (right column) local cloud feedback parameter during progressive 20-year periods between 1970 and 2050. In all panels, stippling indicates regions where the correlation is not significant ($p > 0.05$, two-sided).

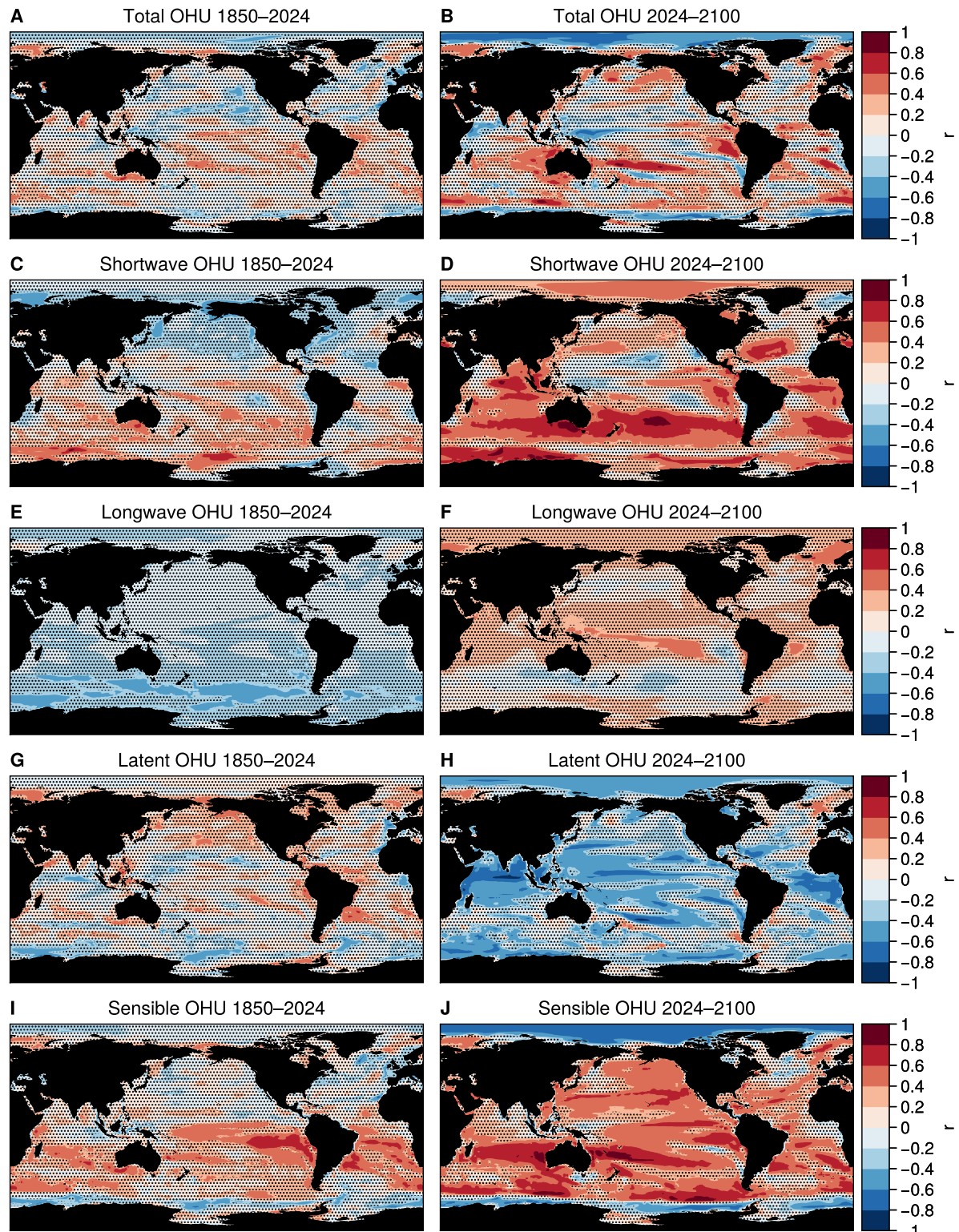

**Figure A5. Relationship between sea ice loss and historical and future OHU components.** Left column: Inter-model correlation between total Antarctic summer sea ice loss and historical 1850–2024 total OHU **(a)** as well as OHU from shortwave **(c)**, longwave **(e)**, latent **(g)**, and sensible heat fluxes **(i)**. Right column: As left column, but for the future 2024–2100 period. In all panels, stippling indicates regions where

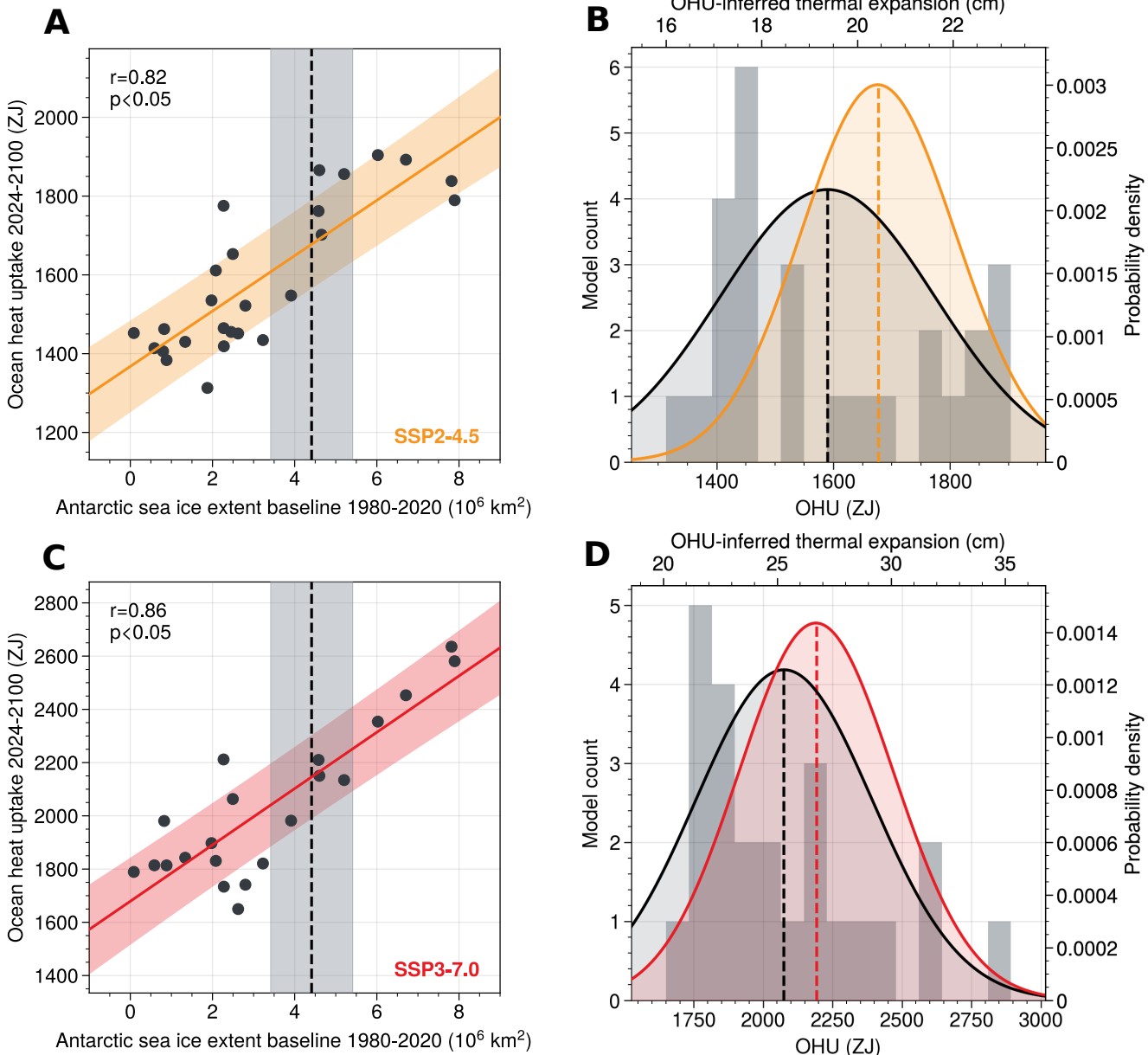

**Figure A6. Emergent constraint on future global ocean heat uptake under SSP2-4.5 and SSP3-7.0. a**, Inter-model relationship between 1980-2020 Antarctic summer (January–February) sea ice extent and cumulative global OHU over 2024–2100 under the SSP2-4.5 scenario. The orange line and shading show the least squares linear regression fit and its uncertainty (see Methods), with the Pearson's correlation coefficient $r$ and two-sided $p$-value given in the upper left corner. The dashed vertical line shows satellite observations of Antarctic summer sea ice extent averaged over 1980–2020 OSI SAF (2017) and the grey shading shows the associated uncertainty of $1 \times 10^6$ km$^2$; this relatively large observational uncertainty ensures we derive a conservative emergent constraint (Methods). **b**, Unconstrained prior (black) and constrained posterior (orange) probability density functions of 2024–2100 global OHU. In grey we show the prior histogram for 2024–2100 OHU (Methods). **c**, as panel **a** but for the SSP3-7.0 scenario. **d**, as panel **b** but for the SSP3-7.0 scenario.

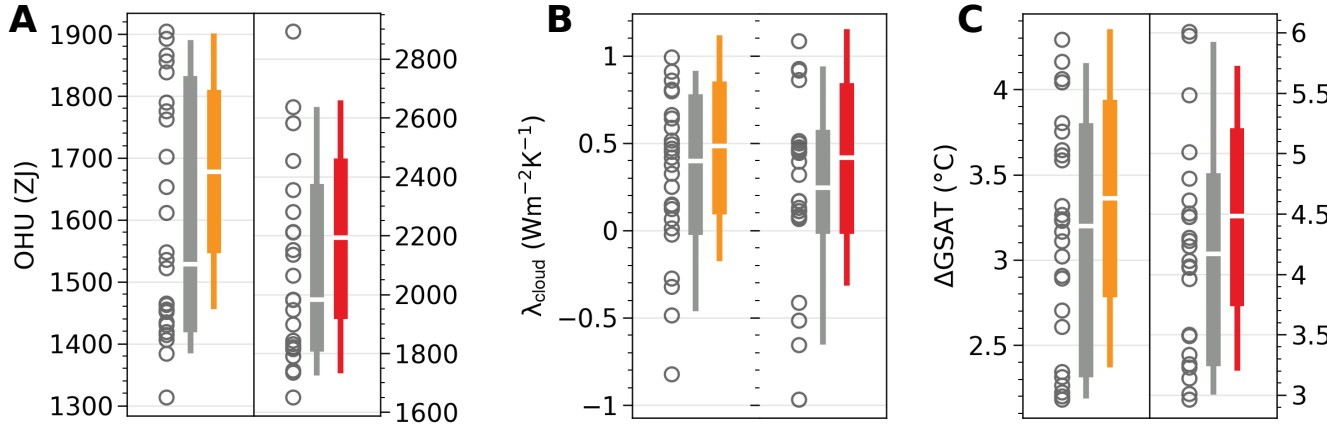

**Figure A7. Constrained distributions of global OHU, cloud feedback, and warming under SSP2-4.5 and SSP3-7.0.** Prior and constrained distributions of **(a)** cumulative global OHU from 2024 to 2100, **(b)** global mean cloud feedback parameter in 2080–2100, and **(c)** global mean surface air temperature (GSAT) anomaly in 2080–2100 relative to the preindustrial. In each panel, distributions are shown for SSP2-4.5 (left) and SSP3-7.0 (right). The grey circles and grey boxplots show the prior distribution of model values, and the yellow and red boxplots show the constrained distributions for SSP2-4.5 and SSP3-7.0, respectively. In each boxplot, the white line shows the median, the central box spans the likely range (66%), and the whiskers extend to a 95% confidence interval. The constrained values are normally distributed by construction (Methods). Note that the $y$-axis scale is different between the two SSPs in panels **a** and **c**.

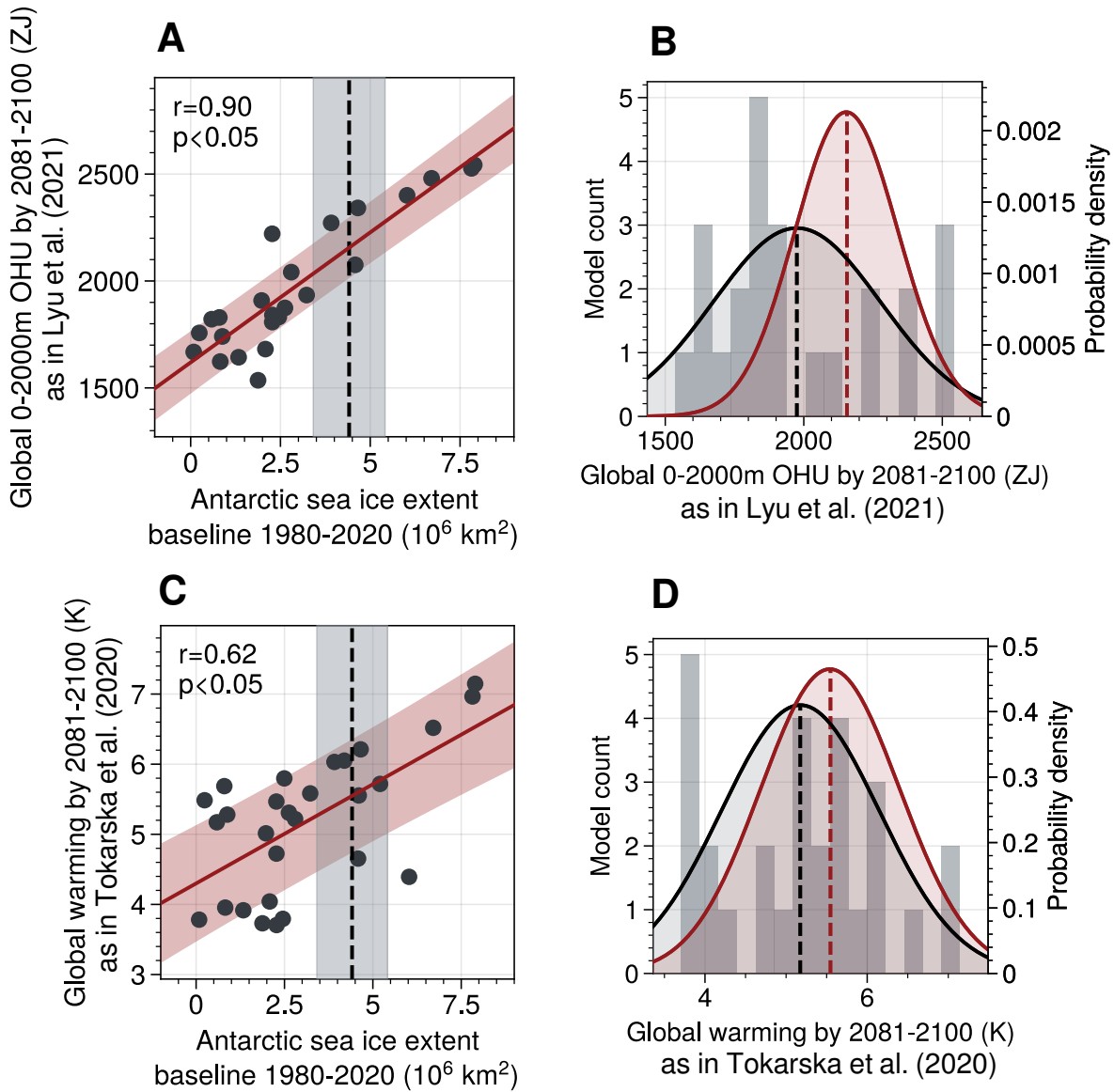

**Figure A8. Emergent constraints on previously published metrics. a**, Inter-model relationship between 1980-2020 Antarctic summer (January–February) sea ice extent and 0–2000m OHU in 2081–2100 relative to 2005–2019 under the SSP5-8.5 scenario (as in Lyu et al. (2021)). The red line and shading show the least squares linear regression fit and its uncertainty (see Methods), with the Pearson's correlation coefficient $r$ and two-sided $p$-value given in the upper left corner. The dashed vertical line shows satellite observations of Antarctic summer sea ice extent averaged over 1980–2020 OSI SAF (2017) and the grey shading shows the associated uncertainty of $1 \times 10^6 \, \text{km}^2$. **b**, Unconstrained prior (black) and constrained posterior (red) probability density functions of 0–2000m global OHU. In grey we show the prior histogram for 0–2000m OHU. **c**, as panel **a** but for global mean atmospheric surface warming in 2081–2100 relative to 1850–1900 under the SSP5-8.5 scenario (as in Tokarska et al. (2020)). **d**, as panel **b** but for global mean atmospheric surface warming as in panel **c**.

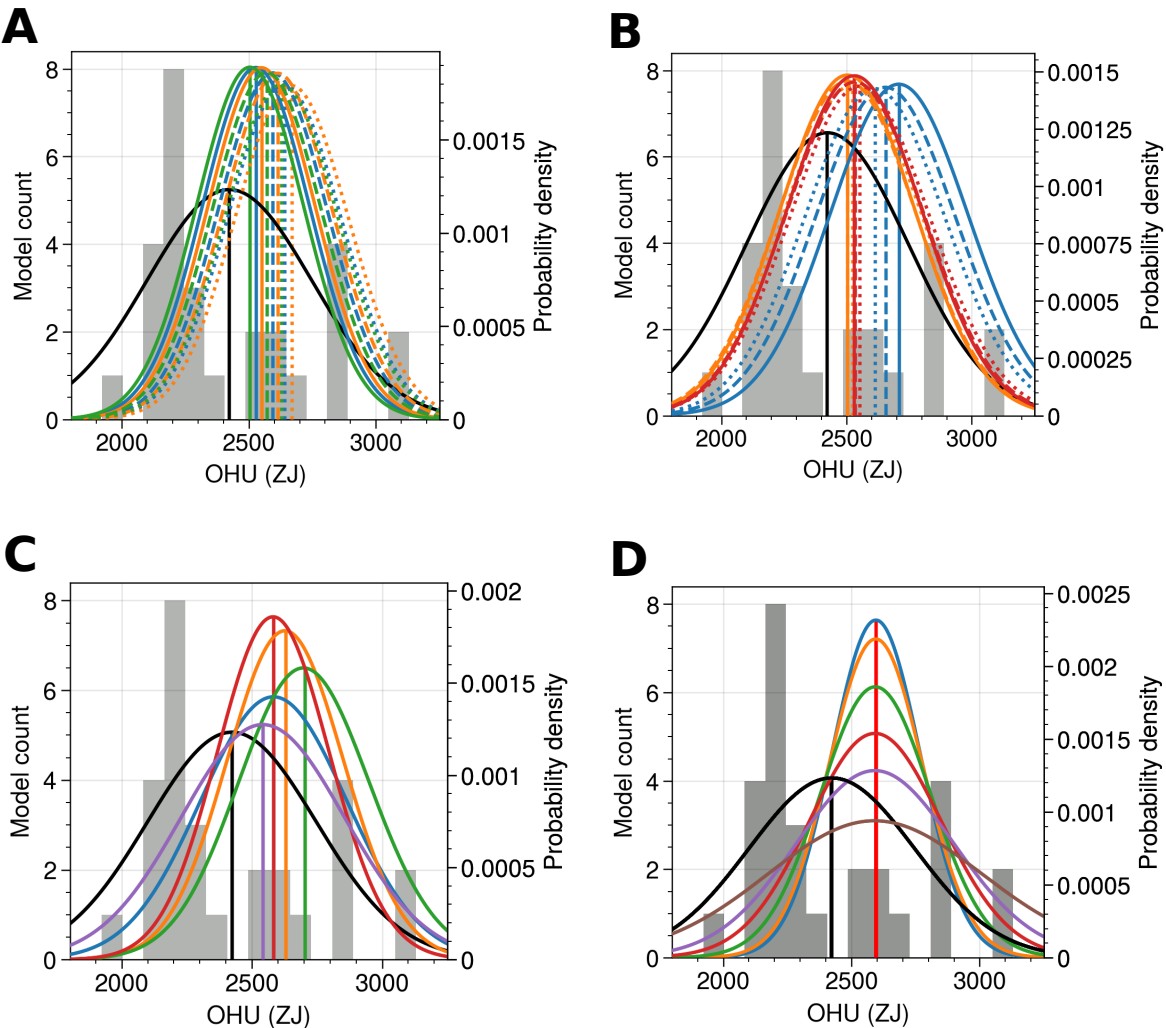

**Figure A9. Robustness of emergent constraint to parameter choices.** Prior OHU histograms and probability density functions (PDFs) as in Fig. 7, and posterior PDFs obtained from different parameter choices. **a**, Different satellite January–February sea ice extent observation sources: OSI SAF (blue), Bootstrap (orange), and NASA Team (green) using different time periods (solid: 1980–2000, dashed: 1990–2010, dotted: 2000–2020). **b**, Different pre-satellite era yearly sea ice extent observation sources: HadISST2.2 (blue), and reconstructions from Dalaiden et al. (2023) (red) and Fogt et al. (2022) (orange), using different time periods (solid: 1920–1960, dashed: 1940–1980, dotted: 1960–2000). **c**, Different season definitions for sea ice extent baseline from the OSI SAF satellite product OSI SAF (2017): yearly (blue), January-February–March (orange), February-March (green), January-February (red), July-August-September (purple). **d**, Different observational uncertainties for January–February sea ice extent from the OSI SAF satellite product: $0.2 \times 10^6$ km$^2$ (blue), $0.5 \times 10^6$ km$^2$ (orange), $1 \times 10^6$ km$^2$ (green), $1.5 \times 10^6$ km$^2$ (red), $2 \times 10^6$ km$^2$ (purple), $3 \times 10^6$ km$^2$ (brown).

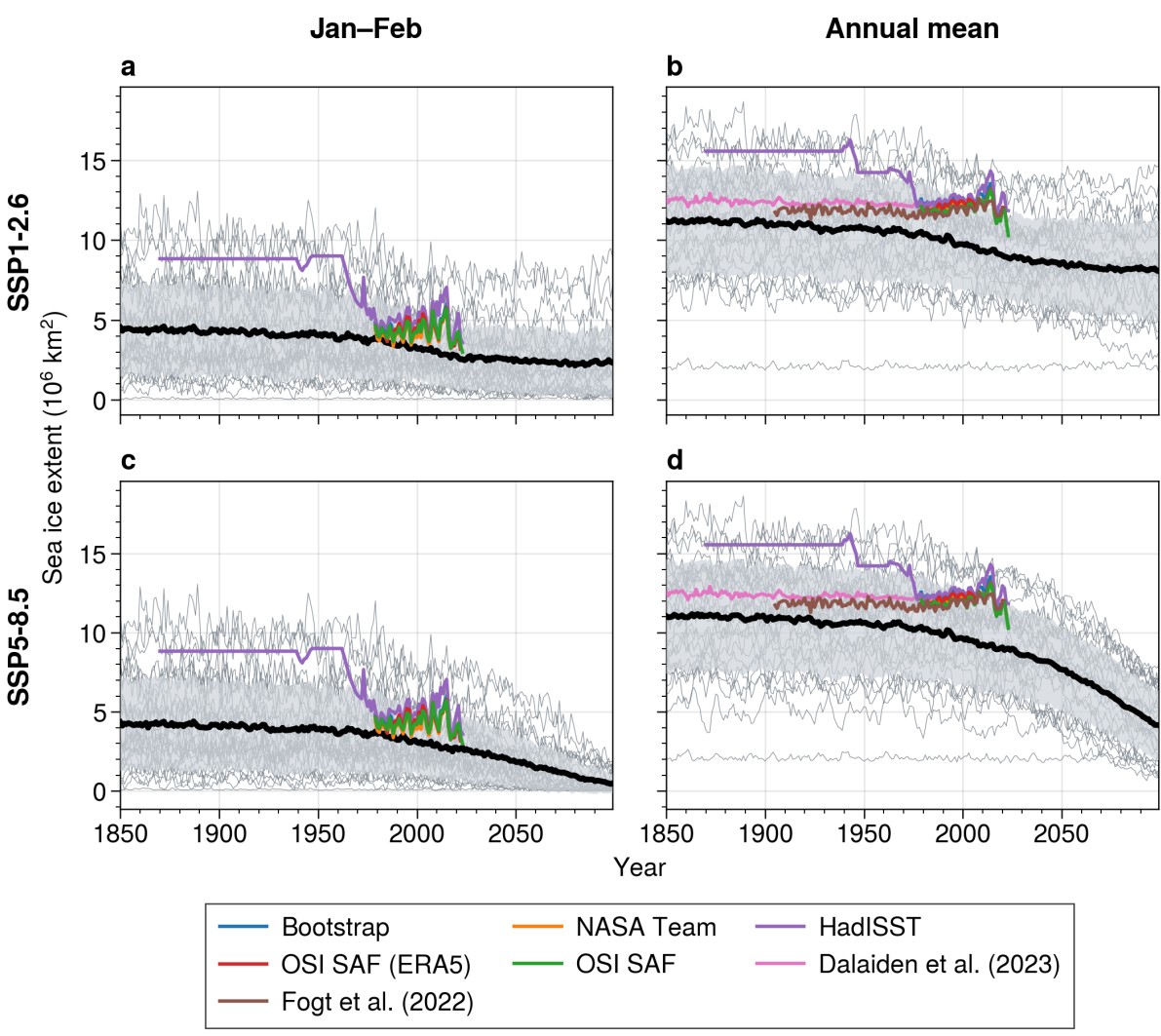

**Figure A10. Time series of observed and simulated Antarctic sea ice extent.** Antarctic sea ice extent simulated by individual CMIP6 models (thin grey lines) and in the ensemble mean (bold black line), and in observational products (colored lines). Model time series extend to 2100 under SSP1-2.6 (**a,b**) and SSP5-8.5 (**c,d**). Yearly values are calculated for (left column) January–February, and (right column) the annual mean.

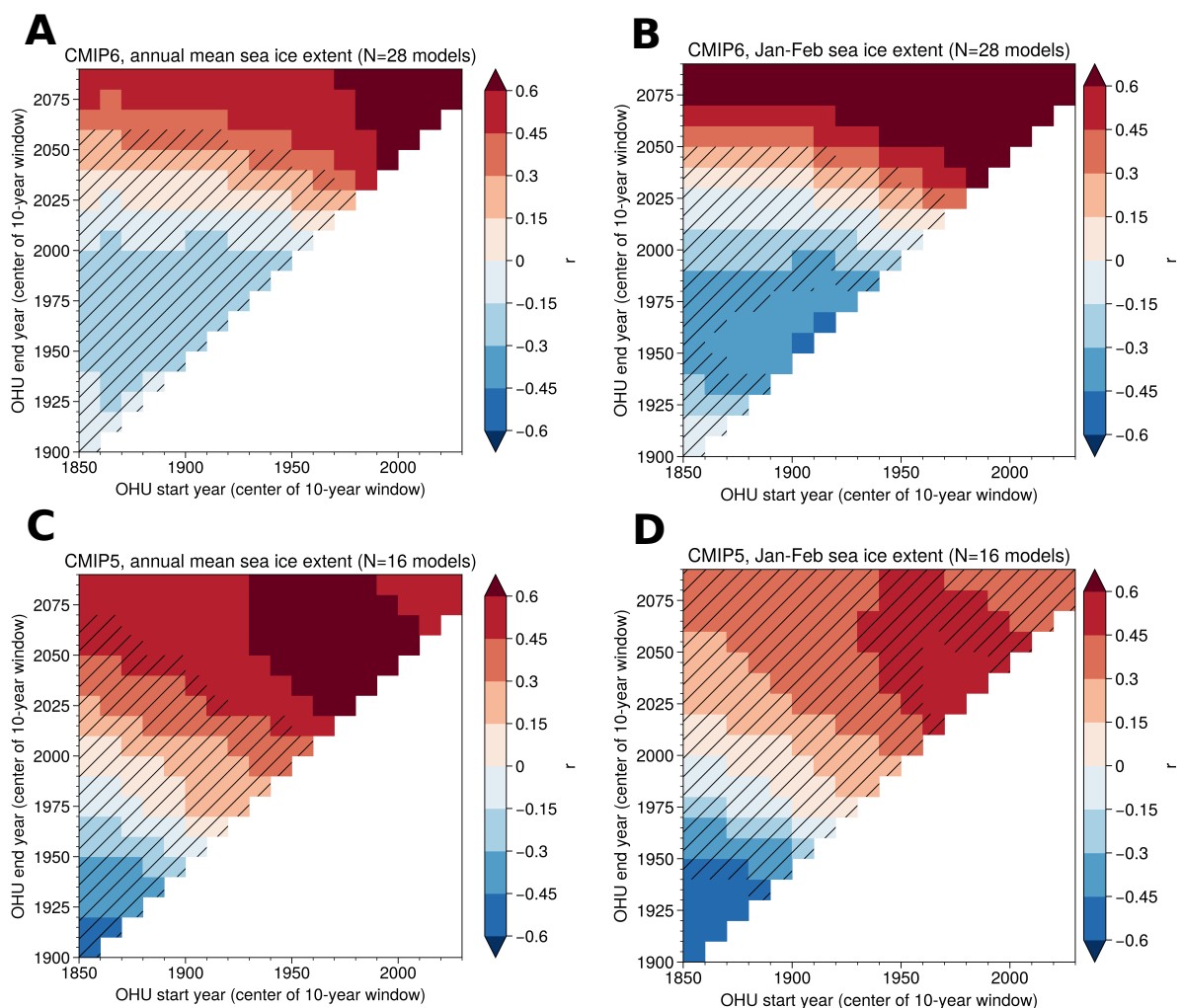

**Figure A11. Sea ice – OHU correlation in CMIP5 and CMIP6 for different values of OHU time period.** Heatmaps of the correlation coefficient between 1980–2020 Antarctic (left column) annual or (right column) January–February sea ice extent and global OHU in (**a**–**b**) CMIP5 under RCP8.5 forcing and (**c**–**d**) CMIP6 under SSP5-8.5 forcing, for different OHU time periods. Stippling indicates parameter values where the sea ice – OHU correlation is not statistically significant ($p >= 0.05$, two-sided).

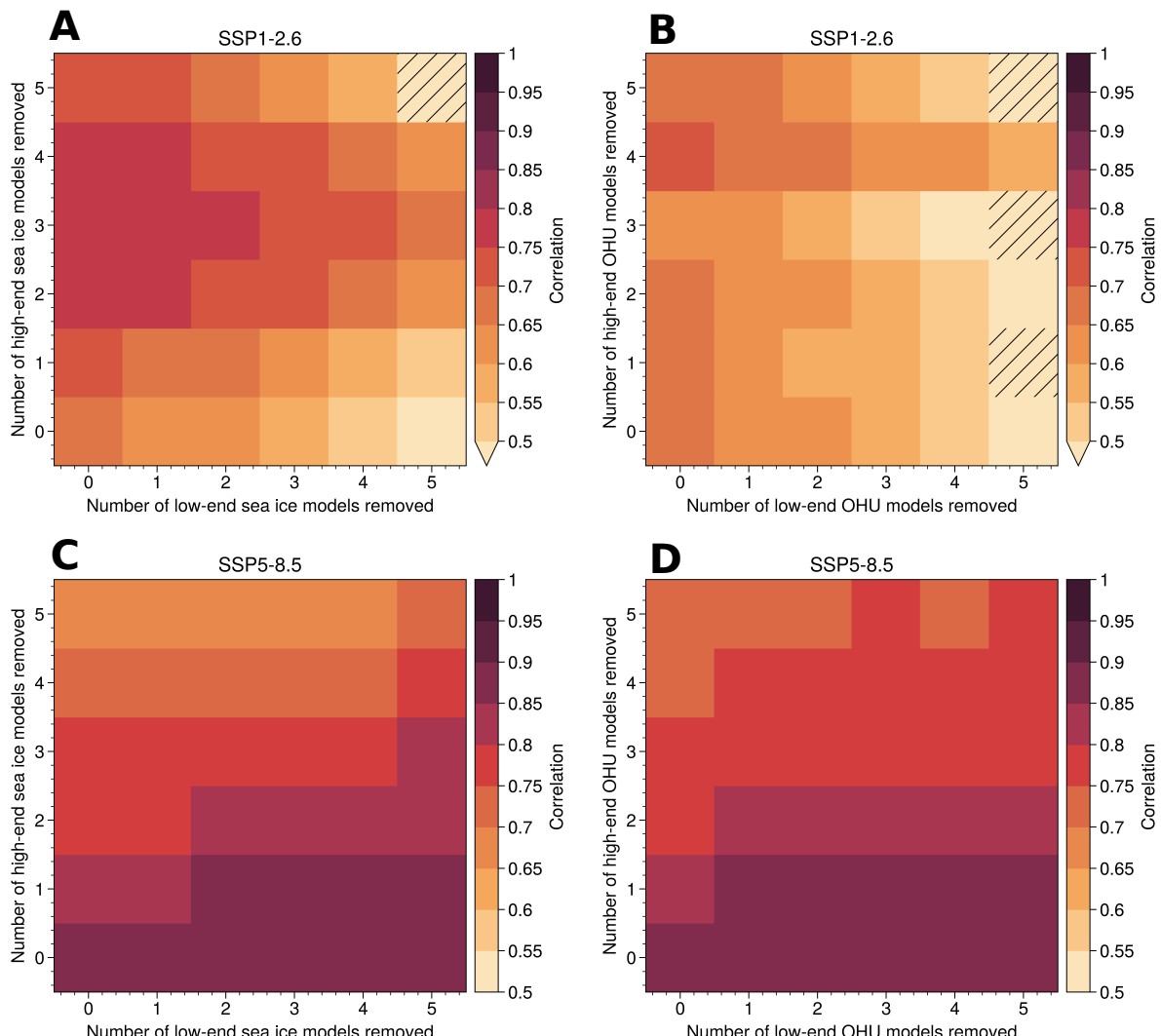

**Figure A12. Robustness of sea ice – OHU correlation to removing extreme model values.** Heatmaps of the correlation coefficient between 1980–2020 Antarctic summer sea ice extent and future (2024–2100) global OHU under (**a–b**) SSP1-2.6 and (**c–d**) SSP5-8.5 when removing a number of models with the highest or lowest sea ice extent (left column), and the highest or lowest future OHU (right column). Stippling indicates parameter values where the sea ice – OHU correlation is not statistically significant ($p >= 0.05$, two-sided).

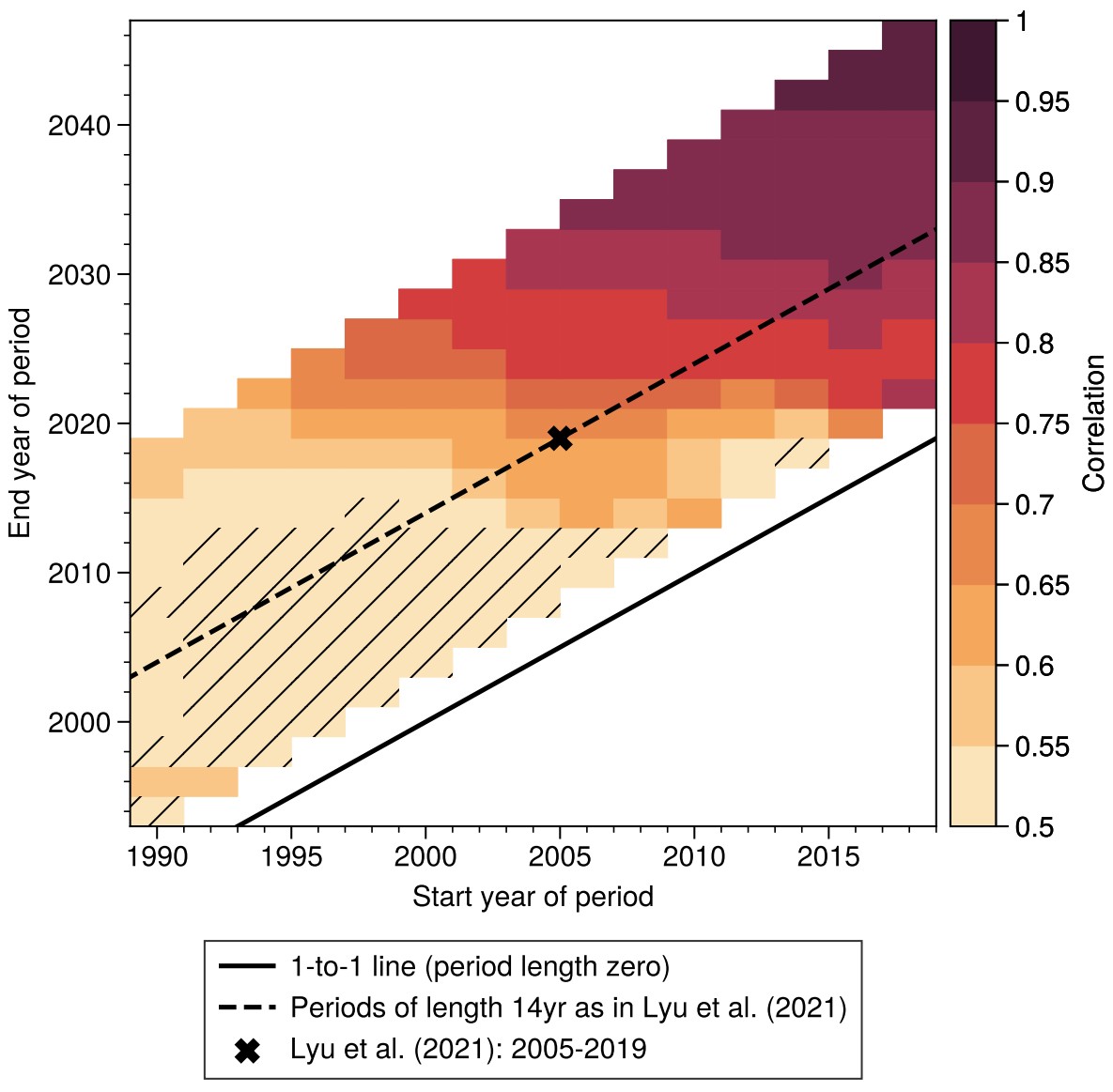

**Figure A13. Sensitivity of OHU constraint based on past warming.** Heatmap of the correlation coefficient between past OHU and future (2081–2100 vs. past) OHU among CMIP6 models under SSP5-8.5 forcing for different choices of the start and end year of the past time period.

**Table A1. CMIP6 models used in this study.** [1]Models for which essential output variables (`hfds` and `siconc`) are unavailable for any of the SSP1-2.6, SSP2-4.5, or SSP3-7.0 scenarios.

| Model | Modeling center | Reference | missing SSPs[1] |
|---|---|---|---|
| CanESM5 | CCCma | Swart et al. (2019) | — |
| CanESM5-CanOE[2] | | | — |
| CMCC-CM2-SR5 | CMCC | Cherchi et al. (2019) | — |
| CMCC-ESM2 | | Lovato et al. (2022) | — |
| CNRM-CM6-1 | CNRM-CERFACS | Voldoire et al. (2019) | — |
| CNRM-CM6-1-HR | | | — |
| CNRM-ESM2-1 | | Séférian et al. (2019) | — |
| ACCESS-ESM1-5 | CSIRO | Ziehn et al. (2020) | — |
| ACCESS-CM2 | CSIRO-ARCCSS | Bi et al. (2020) | — |
| EC-Earth3 | EC-Earth-Consortium | Döscher et al. (2022) | — |
| EC-Earth3-CC | | | ssp126, ssp245, ssp370 |
| EC-Earth3-Veg | | | — |
| EC-Earth3-Veg-LR | | | — |
| IPSL-CM6A-LR | IPSL | Boucher et al. (2020) | — |
| MIROC6 | MIROC | Tatebe et al. (2019) | — |
| HadGEM3-GC31-LL | MOHC | Andrews et al. (2020) | ssp370 |
| HadGEM3-GC31-MM | | | ssp245, ssp370 |
| UKESM1-0-LL | | Sellar et al. (2019) | — |
| MPI-ESM1-2-HR | MPI-M | Gutjahr et al. (2019) | — |
| MPI-ESM1-2-LR | | | — |
| MRI-ESM2-0 | MRI | Yukimoto et al. (2019) | — |
| GISS-E2-1-G | NASA-GISS | Kelley et al. (2020) | — |
| CESM2 | NCAR | Danabasoglu et al. (2020) | — |
| CESM2-WACCM | | | — |
| NorESM2-LM | NCC | Seland et al. (2020) | — |
| NorESM2-MM | | | — |
| GFDL-CM4 | NOAA-GFDL | Held et al. (2019) | ssp126, ssp370 |
| GFDL-ESM4 | | Dunne et al. (2020) | ssp126 |

**Table A2. Emergent constraints across scenarios.** For each variable and each SSP, this table gives the inter-model correlation (Pearson's $r$-value) between 1980–2020 Antarctic summer sea ice extent (SIE) and the respective future variable ($X$), as well as the unconstrained and constrained median values of $X$. Correlation $r$-values with an asterisk indicate significant correlations at the $p < 0.05$ level according to a two-sided Student's $t$-test. Constrained values with an asterisk indicate significant difference between unconstrained and constrained mean values at the $p < 0.05$ level according to a two-sided Student's $t$-test. Uncertainty ranges express the 66% *likely* range. Variable abbreviations stand for ocean heat uptake (OHU), global mean sea level rise from thermal expansion (SLR), global mean surface air temperature warming ($\Delta$GSAT), and global mean cloud feedback parameter ($\lambda_{\mathrm{cloud}}$); see Methods.

| Future scenario | | OHU (ZJ) | SLR (cm) | $\Delta$GSAT (°C) | $\lambda_{\mathrm{cloud}}$ ($\mathrm{W\,m^{-2}\,K^{-1}}$) |
|---|---|---|---|---|---|
| SSP1-2.6 ($n = 25$) | corr($X$, SIE) | $r = 0.66^*$ | $r = 0.66^*$ | $r = 0.45^*$ | $r = 0.64^*$ |
| | prior | $1205 \pm 163$ | $14.7 \pm 2.0$ | $2.25 \pm 0.52$ | $0.43 \pm 0.47$ |
| | constrained | $1244 \pm 141$ | $15.2 \pm 1.7$ | $2.36 \pm 0.51$ | $0.51 \pm 0.41$ |
| SSP2-4.5 ($n = 26$) | corr($X$, SIE) | $r = 0.82^*$ | $r = 0.82^*$ | $r = 0.56^*$ | $r = 0.66^*$ |
| | prior | $1528 \pm 178$ | $18.6 \pm 2.2$ | $3.20 \pm 0.63$ | $0.40 \pm 0.43$ |
| | constrained | $1678^* \pm 129$ | $20.5^* \pm 1.6$ | $3.36 \pm 0.58$ | $0.48 \pm 0.37$ |
| SSP3-7.0 ($n = 24$) | corr($X$, SIE) | $r = 0.64^*$ | $r = 0.64^*$ | $r = 0.62^*$ | $r = 0.63^*$ |
| | prior | $1981 \pm 308$ | $24.2 \pm 3.8$ | $4.17 \pm 0.82$ | $0.24 \pm 0.48$ |
| | constrained | $2193 \pm 270$ | $26.7 \pm 3.3$ | $4.48 \pm 0.73$ | $0.42 \pm 0.42$ |
| SSP5-8.5 ($n = 28$) | corr($X$, SIE) | $r = 0.87^*$ | $r = 0.87^*$ | $r = 0.61^*$ | $r = 0.71^*$ |
| | prior | $2273 \pm 314$ | $27.7 \pm 3.8$ | $5.36 \pm 0.93$ | $0.48 \pm 0.42$ |
| | constrained | $2595^* \pm 208$ | $31.6^* \pm 2.5$ | $5.52 \pm 0.83$ | $0.63^* \pm 0.35$ |

*Author contributions.* Conceptualization: JT, LV, TLF, LK

Methodology: LV, JT

Investigation: all authors

Visualization: LV

Supervision: JT, JBS, CdL

Funding acquisition: JBS, JT

Project administration: JT

Writing—original draft: LV

Writing—review & editing: all authors

*Competing interests.* The authors declare that they have no competing interests.

*Acknowledgements.* The authors thank Juliette Mignot and Ric Williams for discussions, as well as Martin Vancoppenolle, Ted Maksym and Kenza Himmich for help with sea ice extent data.

The authors acknowledge the following funding sources:

Swiss National Science Foundation Ambizione project ArcticECO - PZ00P2_209044 (JT)

Woods Hole Oceanographic Institution postdoctoral scholarship (JT, LV)

Woods Hole Oceanographic Institution guest student program (LV)

EU Horizon 2020 research and innovation programme, grant agreement no. 101137673 (TipESM) (LK, TLF)

ENS Chanel research chair (LK)

EU Horizon 2020 research and innovation programme, grant agreement no. 821001 (SO-CHIC) (JBS, CdL, LV)

To access and process the model outputs and observational data, this study benefited from the IPSL Data and Computing Center ESPRI which is supported by CNRS, SU, CNES and Ecole Polytechnique. We acknowledge the World Climate Research Programme, which, through its Working Group on Coupled Modelling, coordinated and promoted CMIP6. We thank the climate modeling groups for producing and making available their model output, the Earth System Grid Federation (ESGF) for archiving the data and providing access, and the multiple funding agencies who support CMIP6 and ESGF.

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
