# Peer review of "Increased future ocean heat uptake constrained by Antarctic sea ice extent"

_EGUsphere, 2025_

## Author Comment (AC1)

**1 Response to reviewer 1**

**General Comments**

This manuscript presents a new emergent constraint relating present-day Antarctic sea ice extent with future ocean heat uptake in the Southern Ocean. Using this new relationship, an updated estimate of ocean heat uptake is found to be higher than the ensemble mean estimate, and that the uncertainty is greatly reduced. While the increased heat uptake estimate is at odds with previous literature using other emergent constraints, the present-day sea ice extent used here is argued to be a more reliable predictor.

While the novelty of the study might not be obvious at first, since it relies on emergent constraints to reduce uncertainty on an already investigated metric (namely future Southern Ocean heat uptake), this manuscript actually provides a very valuable contribution to the overall understanding of the future of the Southern Ocean. The use of a new predictor, Antarctic sea ice extent, provides a more robust relationship and the new estimate of Southern Ocean heat uptake, of opposite direction compared to previous studies, feeds the discussion for future global and regional climates.

The manuscript is well written but more importantly, the study is well conducted, with several complementary data sources and methodological approaches used to provide a robust conclusion, and all results properly discussed. Many loose ends are convincingly tied in this study, except maybe the use of other sea ice extent-related metrics (especially the seasonal amplitude; see specific comment #3). This makes it a valuable manuscript, and I recommend publication once some minor concerns and modifications have been addressed.

**Response:**

We thank the reviewer for their positive assessment and helpful comments. We have updated the manuscript according to these comments and provide specific responses to each point below.
* * *
**Comment 1.1**

While the cloud feedback is an important process well included and discussed in the manuscript, a short description or definition is lacking for readers less familiar with such a process. This could fit well either in the second paragraph of the introduction or in the short 2.3 section, along with a brief summary of the radiative kernel method.
* * *
**Response:** In response to this suggestion and to comment 2.4 by reviewer #2, we have rewritten and expanded section 2.3 to explain the concept of cloud feedback and the radiative kernel method in more detail:

"Climate feedback parameters (units: $\mathrm{W\,m^{-2}\,K^{-1}}$) quantify the strength of climate feedbacks that either amplify or dampen the climate system's temperature response to radiative forcing (e.g., Ceppi et al., (2017). Among various feedback components such as surface albedo or lapse rate feedback, the cloud feedback is of particular importance due to its large uncertainty (Zelinka et al., 2020). Cloud feedback arises due to changes in a number of cloud properties including cloud amount, altitude, and optical depth. For the quantification of cloud feedback in this study, we compute spatially resolved climate feedback parameters under the SSP5-8.5 scenario using the radiative kernel method (Soden and Held, 2006) with kernels based on the ERA5 reanalysis (Huang and Huang, 2023). The kernel method consists of systematically applying perturbations in variables of interest (such as temperature, humidity, or albedo) in the radiation code of an atmospheric model and diagnosing the resulting change in shortwave and longwave radiation (Soden et al., 2008).

For each variable $X$ (specifically: temperature, water vapor, and surface albedo), this procedure yields a kernel $K_X$ such that

$$\Delta R_X = K_X \cdot \Delta X, \tag{1}$$

where $R_X$ (in $\mathrm{W\,m^{-2}}$) is the radiative response for variable X with anomaly $\Delta X$ (Huang and Huang, 2023). From this, the climate feedback parameter for variable $X$ can be calculated as $\lambda_X = \Delta R_X / \Delta T$, where $\Delta T$ is the global mean surface temperature anomaly.

The cloud feedback parameter is a special case and can not be directly computed from radiative kernels. Instead, it is computed as a residual of all other terms via

$$\Delta R_{\mathrm{cloud}} = \Delta R - \sum_X \Delta R_X - \mathrm{res}^0, \tag{2}$$

where $\Delta R$ is the total radiative response, and

$$\mathrm{res}^0 = \Delta R^0 - \sum_X \Delta R_X^0 \tag{3}$$

is the residual radiative response under clear sky conditions indicated by the superscript 0 (Huang and Huang, 2023)."
* * *
**Comment 1.2**

Alternative predictors: this paragraph highlights that the authors thought through about other predictors to strengthen their results. Yet, there is no clear mention of other ice-related predictors, such as austral winter sea ice extent or amplitude of the seasonal cycle of sea ice extent. CMIP models are known to show discrepancy for both in the Antarctic. Contrarily to the summer extent, winter extent is less constrained by geographical boundaries (opposite of the Arctic, where the winter extent is the one most constrained by land) and might therefore provide a better insight on the potential evolution of OHU. The seasonal amplitude is also likely to influence ocean heat uptake through similar processes as mentioned in the manuscript. It might be worth exploring, but I also strongly suspect that while using those metrics might change some of the quantitative aspects, it won't modify the qualitative story of the study. I therefore leave to the authors to evaluate whether such a new analysis is worth the effort.
* * *
**Response:** We thank the reviewer for this suggestion. In fact, austral winter sea ice extent (defined here as July–August–September) is already analyzed as an alternative predictor as part of the robustness tests in Fig. A9 (the purple line in panel c). This was not yet explicitly mentioned in section 3.3, we have now updated this sentence:

"Using annual mean or austral winter sea ice extent or different definitions of the summer season also yield broadly consistent uncertainty reductions (from $-13\%$ to $-38\%$) and OHU increases (from $+3\%$ to $+11\%$) under SSP5-8.5 (Fig. A9c)".

In general, the inter-model correlation between winter sea ice extent and OHU is slightly lower than between summer sea ice extent and OHU, but the results are qualitatively fully robust to the choice of season, as shown in Fig. A9.

As suggested by the reviewer, we also look at the seasonal amplitude of Antarctic sea ice extent as a potential alternative predictor for future global OHU. A scatter plot between these two quantities across the CMIP6 ensemble is shown below in Fig. R1. We find no significant correlation between sea ice seasonal amplitude and future OHU. As this does not have an impact on the results of our study, we refrain from including this in the main manuscript.

[Figure]

Figure R1: **Relationship between historical Antarctic sea ice extent seasonal amplitude and future global OHU.** The seasonal amplitude is calculated as the difference between winter (JJA) and summer (DJF) average Antarctic sea ice extent for the 1980–2020 period.
* * *
**Comment 1.3**

l.336: As currently worded, this sentence is arguable: the CMIP ensemble mean does not reproduce the observed increasing sea ice trend until 2015. Instead, as shown by Liu (2025), some subset of the ensemble does pick up this increase, but the related members remain a minority (24 versus 53 simulating a decrease). The Liu (2025) study rather highlights the role of internal variability in the present-day sea ice extent trend and the capability of those models to capture this internal variability. Whether this lends confidence to the use of the CMIP5/6 ensemble for emergent constraints based on the pre-industrial or historical mean state is not obvious to me. I suggest removing this sentence, which does not bring much to the discussion, or else clarifying your argument.

**Response:** As suggested by the reviewer, we have now removed this sentence from the manuscript.
* * *
**Comment 1.4**

Figure 8 is poorly inserted in the discussion. This schematic is pleasing to look at and conveys an important summary of the results, but it is not described nor discussed in the text. Please link it better with the main text.

**Response:** Following the suggestion to better integrate the schematic into the text, we have now moved it up into the results section. The schematic is thus now Figure 6, and is explicitly described in the main text:

"This idea is schematically illustrated in Fig.6 which shows how the amplitude of climate warming is preconditioned by the initial climate state."

In addition, it is now also referred to in the fourth and fifth paragraphs of the Discussion.
* * *
**Comment 1.5**

l. 352-359: this paragraph is not clear to me. Do your results invalidate previous results or not? You explain that it does not, but the following sentence still mentions that we should expect larger future warming, while previous studies rather mentioned smaller future warming. Please reformulate and clarify.
* * *
**Response:** The intention in this paragraph was to communicate that the estimates of our emergent constraint are not the "final word" on future OHU projections, since other mechanisms than the one we identify may also be at play and may bias OHU projections in other ways. However, this was formulated in an unclear way, especially the comparison to previous studies. We have now attempted to make this paragraph clearer:

"Our results suggest more warming and heat uptake than the CMIP6 multi-model mean, in contrast to previous studies (Tokarska et al., 2020; Lyu et al., 2021; Nijsse et al., 2020). Our results thus confirm the recent finding that these studies may have underestimated future warming and that the very low equilibrium climate sensitivity (ECS) estimates of some climate models are unlikely (Myhre et al., 2025). Nevertheless, our results do not invalidate previous results indicating that that the extreme end of strong warming and cloud feedback of high-ECS CMIP6 projections is unlikely (Tokarska et al., 2020; Lyu et al., 2021; Nijsse et al., 2020; Jiménez-de-la Cuesta and Mauritsen, 2019; Myers et al., 2021; Cesana and Del Genio, 2021). Furthermore, other shared biases in CMIP6 models could potentially imply additional positive or negative corrections to future OHU projections (e.g., Wang et al., 2024). Ideally, our findings could be combined with other mechanisms impacting future OHU to arrive at a combined observational constraint (e.g., Bretherton and Caldwell, 2020). Endeavours to identify and correct such biases thus remain of utmost importance."
* * *
**Comment 1.6**

l.91: please clarify units of OHU (J)
* * *
**Response:** The units are now mentioned:

"OHU is defined as the anomalous net air-sea heat flux (CMIP6 variable `hfds`) integrated in space and cumulatively integrated in time, resulting in units of Joules: ..."

**Comment 1.7**

In section 2.4, when describing the maths behind the emergent constraint, a short sentence linking the mathematical terms with the physical terms would have been helpful, to better link the theory with how it is applied. (e.g. here, $N = 28$ CMIP models; one of the predictors $x_i$ is the sea ice extent while one of the response variables $y_i$ is the ocean heat uptake, etc.)

**Response:** We have now updated this section according to the reviewer's suggestion:

"Given $N$ realizations of the response variable $y$ and the predictor variable $x$ as well as their least-squares linear fit $f(x) = a + by$ (in the present case, $N = 28$ climate models provide values for the Antarctic sea ice extent predictor $x$ and the global OHU response $y$), the prediction error is..."

**Comment 1.8**

l.141: missing reference for the uncertainty in satellite product being only half of inter-product spread.

**Response:** The missing reference (Wernecke et al. 2024) is now inserted; it is the same reference as for the previous sentence. As our initial manuscript referenced a 2022 preprint of this study, we now reference the published 2024 version.

**Comment 1.9**

l.368: when talking about thermosteric sea level rise, a reminder of the results buried in the section 3.2 would be welcome (e.g. "more thermosteric level rise, estimated to an extra 0.7cm under SSP1-2.6 and 2.1cm under SSP5-8.5 with reduced associated uncertainties, more damage to marine ecosystems [...]")

**Response:** We now explicitly mention the sea level rise estimates resulting from our ocean heat uptake constraint in this sentence:

"Increased ocean heat uptake would cause more thermosteric sea level rise (our central estimates for total end-of-century sea level rise are between 15–31 cm, depending on scenario), more damage to marine ecosystems and create additional risks to socio-economic systems."

**2   Response to reviewer 2**
* * *
**General Comments**

This study provides improved global ocean heat uptake (OHU) projections by identifying a relationship between present-day Antarctic sea ice extent and future OHU across an ensemble of climate models. Combining this relationship with satellite observations of Antarctic sea ice reduces the uncertainty of OHU projections under future emission scenarios considerably. This emergent constraint is based on a strong coupling between Antarctic sea ice, deep ocean temperatures, and Southern Hemisphere sea surface temperatures and cloud cover in climate models. The robustness of this emergent constraint is thoroughly discussed and comparison with previous constraints based on past warming trends has been done. Overall, this paper is carefully written with detailed analysis and discussion. I only have some minor comments.
* * *
**Response:**

We thank the reviewer for their positive assessment and helpful comments. We have updated the manuscript according to these comments and provide specific responses to each point below.
* * *
**Comment 2.1**

Line 200: Sea ice loss is mentioned here which is different from the sea ice extent used in Figure 5. For consistency, it would be good to show similar figures using sea ice loss, although a significant correlation between sea ice loss and sea ice extent is identified in Figure 3a.
* * *
**Response:**

We thank the reviewer for this suggestion. We have recreated Figure 5 using Antarctic sea ice extent anomaly in each time period instead of preindustrial sea ice extent; this is shown below in Fig. R2. As noted by the reviewer, the strong correlation between preindustrial sea ice extent and sea ice extent decline over the 21st century (Fig. 3a in the main manuscript) means that Fig. R2 is very similar to the original Fig. 5, only with reversed sign (since the sea ice anomaly is negative). In the main part of the manuscript, all figures currently use preindustrial sea ice extent instead of sea ice loss, and we would preferably keep this for overall consistency. Taking into account the reviewer's comment, we thus propose to improve the consistency in this paragraph by instead rewording the text to refer to preindustrial sea ice extent instead of sea ice loss, which also describes Fig. 5 more accurately.

We have thus adapted the beginning of this paragraph in the following way (italics show the changes):

"The cloud feedback connects *preindustrial* Antarctic sea ice extent and future global OHU. Across the ESM ensemble, this connection is globally apparent by the end of the 21st century as strong correlations between cloud feedback and Antarctic sea ice extent loss (Fig.A3a), and between cloud feedback and global OHU (Fig.A3b). The global extent of this relationship between Antarctic sea ice loss, cloud feedback and OHU is the result of a northward propagation of this relationship originating in the Southern Ocean. The surface warming signal in the ocean and atmosphere related to *preindustrial* sea ice extent first emerges..."

[Figure]

Figure R2: **Time evolution of sea ice loss–related surface warming and cloud feedback.** Inter-model correlation across CMIP6 models under SSP5-8.5 between Antarctic summer sea ice extent anomaly and (top row) local sea surface temperature anomaly, (bottom row) local total cloud feedback parameter $\lambda_{\mathrm{cloud}}$, during different 20-year periods between 1970 and 2100. In all panels, stippling indicates regions where the correlation is not significant ($p \geq 0.05$, two-sided).
* * *
**Comment 2.2**

Line 259: As described in the Methods section, OHU is defined as the anomalous net air-sea heat flux integrated in space and cumulatively integrated in time, which is different from temperature. Does the region where there is significant correlation between zonal mean ocean warming and preindustrial sea ice extent in Figure 4 necessarily show "addition OHU"?
* * *
**Response:** The OHU calculated by integrating air-sea heat flux should agree with the OHU obtained from 3D ocean temperature on the native model grid due to conservation of energy. Therefore, we do expect the zonal mean ocean warming in Figure 4 to correspond to exactly the same "kind" of OHU as used everywhere else in the study.
* * *
**Comment 2.3**

I wonder why some figures (e.g. Figure 2, Figure 5) show thinner stippling while some other figures (e.g. Figure 4) show much thicker stippling. It may not be due to different grid resolution, I guess. It would be good to use the same stippling across different figures in the whole paper.
* * *
**Response:** The stippling pattern used for all figures is nominally the same, and I am unfortunately unaware of how to fine-tune the spacing of stippling points given the different grid resolutions as the reviewer mentions.

> **Comment 2.4**
>
> Section 2.3: Could you provide more details on how the climate feedback parameters are computed? For example, showing some major equations would be helpful.

**Response:** In response to this suggestion and to comment 1.1 by reviewer #1, we have rewritten and expanded section 2.3 to explain the concept of cloud feedback and the radiative kernel method in more detail, including the most important equations for the kernel method:

"Climate feedback parameters (units: $\mathrm{W\,m^{-2}\,K^{-1}}$) quantify the strength of climate feedbacks that either amplify or dampen the climate system's temperature response to radiative forcing (e.g., Ceppi et al., (2017). Among various feedback components such as surface albedo or lapse rate feedback, the cloud feedback is of particular importance due to its large uncertainty (Zelinka et al., 2020). Cloud feedback arises due to changes in a number of cloud properties including cloud amount, altitude, and optical depth. For the quantification of cloud feedback in this study, we compute spatially resolved climate feedback parameters under the SSP5-8.5 scenario using the radiative kernel method (Soden and Held, 2006) with kernels based on the ERA5 reanalysis (Huang and Huang, 2023). The kernel method consists of systematically applying perturbations in variables of interest (such as temperature, humidity, or albedo) in the radiation code of an atmospheric model and diagnosing the resulting change in shortwave and longwave radiation (Soden et al., 2008).

For each variable $X$ (specifically: temperature, water vapor, and surface albedo), this procedure yields a kernel $K_X$ such that

$$\Delta R_X = K_X \cdot \Delta X, \tag{4}$$

where $R_X$ (in $\mathrm{W\,m^{-2}}$) is the radiative response for variable X with anomaly $\Delta X$ (Huang and Huang, 2023). From this, the climate feedback parameter for variable $X$ can be calculated as $\lambda_X = \Delta R_X / \Delta T$, where $\Delta T$ is the global mean surface temperature anomaly.

The cloud feedback parameter is a special case and can not be directly computed from radiative kernels. Instead, it is computed as a residual of all other terms via

$$\Delta R_{\mathrm{cloud}} = \Delta R - \sum_X \Delta R_X - \mathrm{res}^0, \tag{5}$$

where $\Delta R$ is the total radiative response, and

$$\mathrm{res}^0 = \Delta R^0 - \sum_X \Delta R_X^0 \tag{6}$$

is the residual radiative response under clear sky conditions indicated by the superscript 0 (Huang and Huang, 2023)."